



Atmospheric
Chemistry
and Physics

# Measurement report: Characterization of uncertainties in fluxes and fuel sulfur content from ship emissions in the Baltic Sea

**Jari Walden**[1], **Liisa Pirjola**[2,3], **Tuomas Laurila**[1], **Juha Hatakka**[1], **Heidi Pettersson**[4], **Tuomas Walden**[1], **Jukka-Pekka Jalkanen**[1], **Harri Nordlund**[2], **Toivo Truuts**[5], **Miika Meretoja**[6], and **Kimmo K. Kahma**[4]

[1]Climate Research Program, Finnish Meteorological Institute, Helsinki, Finland
[2]Department of Automotive and Mechanical Engineering, Metropolia University of Applied Sciences, Vantaa, Finland
[3]Aerosol Physics Laboratory, Faculty of Engineering and Natural Sciences, Tampere University, Tampere, Finland
[4]Meteorological and Marine Research Program, Finnish Meteorological Institute, Helsinki, Finland
[5]Air Quality Management Department, Estonian Environmental Research Centre, Tallinn, Estonia
[6]Environmental Protection Unit, City of Turku, Finland

**Correspondence:** Jari Walden (jari.a.walden@gmail.com)

Received: 18 October 2020 – Discussion started: 15 January 202
Revised: 10 September 2021 – Accepted: 19 October 2021 – Published:

**Abstract.** Fluxes of gaseous compounds and nanoparticles were studied using micrometeorological methods at Harmaja in the Baltic Sea. The measurement site was situated beside the ship route to and from the city of Helsinki. The gradient (GR) method was used to measure fluxes of $SO_2$, NO, $NO_2$, $O_3$, $CO_2$, and $N_{tot}$ (the number concentration of nanoparticles). In addition, the flux of $CO_2$ was also measured using the eddy-covariance (EC) method. Distortion of the flow field caused by obstacles around the measurement mast was studied by applying a computation fluid dynamic (CFD) model. This was used to establish the corresponding heights in the undisturbed stream. The wind speed and the turbulent parameters at each of the established heights were then recalculated for the gradient model. The effect of waves on the boundary layer was taken into consideration, as the Monin–Obukhov theory used to calculate the fluxes is not valid in the presence of swell. Uncertainty budgets for the measurement systems were constructed to judge the reliability of the results. No clear fluxes across the air–sea nor the sea–air interface were observed for $SO_2$, NO, $NO_2$, $NO_x$ ($= NO + NO_2$), $O_3$, or $CO_2$ using the GR method. A negative flux was observed for $N_{tot}$, with a median value of $-0.23 \times 10^9$ m$^{-2}$ s$^{-1}$ and an uncertainty range of 31 %–41 %. For $CO_2$, while both positive and negative fluxes were observed, the median value was $-0.0036$ mg m$^{-2}$ s$^{-1}$ TS1 with an uncertainty range of 30 %–60 % for the EC methods. Ship emissions were responsible for the deposition of $N_{tot}$, while they had a minor effect on $CO_2$ deposition. The fuel sulfur content (FSC) of the marine fuel used in ships passing the site was determined from the observed ratio of the $SO_2$ and $CO_2$ concentrations. A typical value of $0.40 \pm 0.06$ % was obtained for the FSC, which is in compliance with the contemporary FSC limit value of 1 % in the Baltic Sea area at the time of measurements. The method to estimate the uncertainty in the FSC was found to be accurate enough for use with the latest regulations, 0.1 % (Baltic Sea area) and 0.5 % (global oceans).

## 1 Introduction

The Baltic Sea, owing to its nature as a relatively small inland sea that experiences heavy ship traffic and is surrounded by populated areas, is very sensitive to pollutants. Due to the very narrow and shallow strait of Kattegat in Denmark, the exchange of seawater between the North Sea and the Baltic Sea is limited. The load of phosphorus and nitrogen in the Baltic Sea mainly comes from rivers. The rivers bring in fresh water, maintaining the salinity of the seawater in the Baltic Sea at 1/10 of that of the ocean. In addition, the airborne deposition of pollutants from the emissions of ships and from industry are becoming more and more important sources (Hongisto and Joffre, 2005). Ship emissions of most air pollutants except CO decreased during the period from 2006 to 2018, but greenhouse gas emissions from

ships remained stable throughout the abovementioned period, regardless of the growth of ship transport reported in tonne kilometers (HELCOM, 2019). Ship emissions enter the sea mostly via indirect deposition of sulfur and nitrogen compounds through chemical conversion in the atmosphere (de Leeuw et al., 2003; Hongisto and Joffre, 2005; Hongisto, 2014), or via direct deposition from the gas phase. In the Baltic Sea, few measurement facilities have been set up to measure the gas exchange between the sea–air interface using micrometeorological methods (Rutgersson et al., 2020; Honkanen et al., 2018). Nevertheless, the need for a reduction of atmospheric pollutants in the emissions has been taken seriously by the International Maritime Organization (IMO), which launched the MARPOL agreement for the reduction of ship emissions, Annex VI (IMO, 1997 TS2). The latest revision includes more stringent emission limits for $NO_x$ and $SO_2$. In spite of these abatement regulations, the ship emissions of IMO-registered vessels and non-IMO-registered vessels show constant or slightly increasing trends in $NO_x$, $SO_2$, and $PM_{2.5}$ compounds as well as a clearly increasing trend in CO. Once the stringent regulation of the fuel sulfur content (FSC) in marine fuel came into power on 1 January 2015, the emissions of $SO_2$ and $PM_{2.5}$ decreased rapidly at both regional and global levels (Johansson and Jalkanen, 2016; Jonson et al., 2020; Seppälä et al., 2021).

The goal of this study was to (i) measure the gas and nanoparticle exchange between the sea–air interface in a marine coastal environment close to ship routes, (ii) study the transport and dispersion of a ship plume to the footprint area, (iii) define the FSC of ship emission plumes, and (iv) characterize the uncertainty sources of the measurement results. The measurements took place in the Baltic Sea on the small island of Harmaja in the vicinity of the city of Helsinki during the summers of 2011 and 2012. The ship routes between the city of Helsinki and the cities of Tallinn, Stockholm, and St. Petersburg pass by the measurement site. The exchange of gaseous $NO_x$, $SO_2$, $CO_2$, $O_{3,}$, and fine particles across the sea and atmosphere surface layer was studied using micrometeorological methods. The fluxes of these compounds were measured using the gradient method and the eddy-covariance method. In addition, the concentration of methane was measured. The major sources of pollutants were ship emissions, but transboundary emissions and emissions from the sea and from the city of Helsinki also contributed to the observed concentration levels of pollutants. The FSC of the ships' marine fuel was determined from the measured concentrations of $SO_2$ and $CO_2$ (Cooper, 2005) in order to determine the compliance of the fuel used with regulations. Ships passing the measurement site were identified using automatic identification system (AIS) data (Jalkanen et al., 2009). This method has been used to define the FSC in previous studies (Alföldy et al., 2013; Moldanová et al., 2013; Pirjola et al., 2014). Recently, it has also been used as an indicator method for routine control by the authorities in some countries (Mellqvist, 2018). The uncertainty of the flux and FSC measure-

ments was estimated based on the performance of the analyzers and measurement probes for meteorological parameters; for the FSC, the uncertainty of defining the peak areas of emission plumes from the ships was also taken into account.

## 2  Theoretical background

### 2.1  Micrometeorological methods

Micrometeorological methods are used to measure gas exchange across the surface layer (Kaimal and Finnigan, 1994); of these, the eddy-covariance (EC) method and the gradient (GR) method are commonly used. The use of micrometeorological methods requires certain criteria to be met with respect to the atmospheric conditions: the homogeneity of the turbulence flow field on the footprint area, the stationarity of the measuring processes, and the absence of swell (e.g., Foken and Wichura 1996, Miller et al 2010, Drennan et al 2003 CE1. As such, in order to ensure the stationarity of the flux measurements, the footprint area and the occurrence of swell were considered. The eddy-covariance method is a direct flux measurement method, whereas the gradient method is an indirect measurement method. In the eddy-covariance method, the flux of a gas compound is measured using fast sensors (response better than 10 Hz) to measure the fluctuation in wind velocities and the concentration of chemical compounds. In contrast, the gradient method overcomes the problem of the fast analysis of chemical compounds. However, this method requires that the atmospheric conditions are stationary, needs very accurate measurements of the parameters that it uses (Businger, 1986), and assumes a constant layer flux (Dyer and Hicks, 1970). In this study, both the GR and the EC methods were used: the gradient method was used for gas compounds (oxides of nitrogen, ozone, sulfur dioxide, and carbon dioxide) and nanoparticles, and the eddy-covariance method was used to measure the flux of carbon dioxide. In the GR method, the wind speed should be measured at different heights, usually with conventional cup anemometers, whereas in the EC method, the fluctuation in the three-dimensional wind speed is measured by a sonic anemometer. Short descriptions of both methods are given below, with more emphasis on the gradient method.

$$F_c = -K_c \frac{\partial c}{\partial z}, \qquad (1)$$

where $F_c$ is the flux of the scalar quantity $c$, and $K_c$ is the eddy diffusivity of $c$; here, $c$ refers to the gas compounds and particles. The gradient $\partial c/\partial z$ describes the mean concentration of $c$ in the vertical direction $z$. By definition, the flux is opposite to the gradient, which is positive towards the increasing concentration. The eddy diffusivity for a chemical compound $c$ is calculated by applying an assumption $K_c = K_h$ (Businger, 1986), i.e., the eddy diffusivity of the gas concentration is the same as that for heat. The eddy dif-

Please note the remarks at the end of the manuscript.

fusivity for heat transfer can be expressed as follows:

$$K_h = \kappa \cdot u_* \cdot z / \phi_h, \tag{2}$$

where $\phi_h$ is the dimensionless temperature gradient. Equation (1) can now be rewritten with the help of Eq. (2) in the following form:

$$F_c = -\frac{\kappa u_* z}{\phi_h} \frac{\partial c}{\partial z}, \tag{3}$$

where $\kappa$ is the von Kármán constant ($\approx 0.4$), $u_*$ is the friction velocity, $c$ is the mean concentration of gases or particulate matter, and the dimensionless temperature gradient $\phi_h$ is defined as

$$\phi_h = \frac{\kappa z}{\theta^*} \frac{\partial \overline{\theta}}{\partial z}. \tag{4}$$

Here, $\theta$ is the potential temperature (Panofsky and Dutton, 1987), and $\theta^* (= -\overline{\theta' w'}/u_*)$ is the scaling parameter for temperature. The dimensionless potential function, Eq. (4), can be written in a way similar for the momentum $\phi_m$ and for the mean concentration of gases or particulate matter $\phi_c$, respectively CE2. The line above the symbol denotes the average of the quantity over time. Equation (3) can be presented in the following form:

$$F_c = -\frac{\kappa u_* (c(z_2) - c(z_1))}{\left( \ln\left(\frac{z_2}{z_1}\right) - \Psi_c(\zeta_2) + \Psi_c(\zeta_1) \right)}, \tag{5}$$

where $\psi_c(\xi_2)$ and $\psi_c(\xi_1)$ are the integral functions of Eq. (4) over the stability parameter $\xi$ at heights $z_2$ and $z_1$, respectively. The dimensionless gradient function Eq. (4) can be represented in the semiempirical form of Businger and Dyer (Businger et al., 1971; Dyer, 1974), which is used here for the calculations. The stability parameter $\xi$ is related to the Monin–Obukhov (M–O) length, $L$, according to the relation $\xi = z/L$, where $z$ is the height from the surface. Equation (5) is valid under neutral ($\xi = 0$) and unstable ($\xi < 1$) conditions but not under stable conditions ($\xi \gg 1$) (Panofsky and Dutton, 1987).

In Eq. (5), the concentration difference at heights $z_2$ and $z_1$ is known from the measurements, but the friction velocity, the integral functions ($\psi_c$), and the stability parameter are unknown. To solve the turbulence parameters, a method proposed by Paulson (1970) was applied.

On the other hand, the sonic anemometer measures the wind velocity with the help of acoustic pulses that propagate along the path between the sound emitter and the receiver. The three-dimensional wind components, i.e., horizontal ($u, v$) and vertical ($w$), are measured based on the changes in the acoustic signals along the fixed path lengths. The momentum flux $F_m$ measured by the sonic anemometer can be calculated from the expression

$$F_m = \rho \overline{u'w'} = \rho u_*^2, \tag{6}$$

where $\rho$ is the air density, $u'$ and $w'$ are the fluctuations in the wind speed components measured by the sonic anemometer, and $u_*$ is the friction velocity. The friction velocity can be calculated from the surface stress according to

$$u_* = \left( \overline{-u'w'} \right)^{1/2}. \tag{7}$$

Once the friction velocity is calculated according to Eq. (7), the Monin–Obukhov length can be expressed as follows:

$$L = -\frac{u_*^3 \theta}{\kappa g Q_{*o}} = -\frac{u_*^3 T}{\kappa g \overline{w'T'}}, \tag{8}$$

where $\theta$ is the potential temperature (in K), $g$ is the acceleration of gravity, and $Q_{*o}$ is the heat flux at the surface. The right side of Eq. (8) is an approximation, where $T$ is the air temperature, and $\overline{w'T'}$ is the heat flux measured by sonic anemometer. The friction velocity and the M–O length obtained from the sonic measurements were used to complete the solutions for the GR method. The M–O similarity theory states that the mean and turbulence variables in the surface layer are functions of height near the ground. General conditions for the M–O similarity theory are a horizontally homogeneous surface structure, stationary (or near-stationary) conditions (e.g., Foken and Wichura, 1996), a constant flux layer, and that the atmospheric turbulence affects the vertical profiles of wind speed, potential temperature, and humidity. In addition to the abovementioned conditions, the M–O similarity theory has been found to be valid over slowly moving waves, which sufficiently resemble solid ground, in the marine environment, but it fails in the presence of swell, i.e., waves that move faster than the wind (e.g., Drennan at al., 1999, 2003; Smedman et al., 1999). The effects of swell on the boundary layer are manifold (Högström et al., 2008, 2013, 2015), with the most conspicuous being the absence of a vertical velocity gradient above a certain wavelength-dependent height. From the results of Kahma et al. (2016), it can be deduced that the absence of wave components that are faster than the wind speed at 10 m height is typically sufficient to ensure that the waves do not invalidate the M–O similarity theory.

Similarly to the momentum flux in Eq. (6), one can express the vertical flux of a gas compound (e.g., $CO_2$) as follows:

$$F_c = \overline{\rho_a} \overline{w'c'}, \tag{9}$$

where $\rho_a$ is the density of dry air, and $c'$ is the measured molar fraction of $CO_2$ ($\mu mol\, mol^{-1}$). The commonly used infrared analyzers normally measure the concentration of $CO_2$ in air under wet conditions unless an air drier is used in the sampling tube. The widely used method to correct the fluctuations in water vapor and heat is the so-called "WPL method" proposed by Webb et al. (1980), which is applied in this study.

## 2.2 NO–O₃–NO₂ chemistry

The chemical interconversion of the NO–O$_3$–NO$_2$ system is well-known and described in the literature (Seinfeld and Pandis, 2012). In the atmosphere, the NO–O$_3$–NO$_2$ reaction system forms a cycle where the reaction forming NO$_2$ (i.e., the reaction NO+O$_3$) is the reverse of the reaction for dissociating NO$_2$ to form NO, which takes place in the presence of sunlight (at wavelengths $<$ 420 nm).

The chemical cycle in the NO–O$_3$–NO$_2$ system is fast, depending on the concentration of the compounds and the available sunlight (day-/night-time). The timescale ($t$) for vertical mixing can be estimated from the relation $t = z/u_*$. With $z$ as 10 m and the friction velocity between 0.1 and 0.5 m s$^{-1}$, the timescale for vertical mixing is 20–100 s. This is of the same order of magnitude as the timescale of the NO–O$_3$–NO$_2$ system. Therefore, the assumption of a vertical constant flux according to M–O theory is not valid. The approximation $K_h = K_c$ is not correct; thus, the turbulent exchange coefficient $K_c$ (Eq. 2) must be modified. This problem has been discussed by several authors (Lenschow and Delany, 1987; Kramm et al., 1991; Vila-Guerau de Arellano et al., 1993; Duyzer et al., 1995). Lenschow and Delany (1987) constructed an analytical formulation for the flux profiles of the NO and NO$_2$ compounds as a function of height, and Duyzer et al. (1995) developed a correction procedure for the formula developed in the abovementioned study.

## 2.3 Estimation of the sulfur content in the marine fuel of ships

The maximum allowed sulfur content in marine fuel used on the oceans is defined in Annex VI of the MARPOL agreement (IMO, 2008). The agreement also defines the sea areas where a lower marine fuel sulfur content must be used. These restricted sea areas, called "SO$_x$ Emission Control Areas" (SECA) need to be approved by the countries in the proposed SECA. The Baltic Sea and the North Sea (IMO, 2008) form a SECA area in which a stringent sulfur limit for the marine fuels applies. In 2012, the European Union (EU) updated the directive to include the more stringent demands from MARPOL Annex VI (Directive 2012/33/EU, 2012 TS3). According to MARPOL Annex VI, the testing of sulfur content must be undertaken in accordance with the International Organization for Standardization (ISO) standard, which includes a set of key tests, e.g., to identify potential fuel issues where exceedances of emission may occur (ISO 8217:2012). According to ISO 8217, the standard method for the assessment of the sulfur content must be in accordance with ISO 8754 (ISO 8754:2003). The main issue here is that the sample for the analysis of the sulfur content needs to be taken from the bunker fuel at the harbor, not during the cruise.

A method that can be used to determine the FSC during a cruise is to measure the ratio of $\Delta SO_2$ to $\Delta CO_2$ from the emissions, either from the stack or from the ambient air. The

$\Delta SO_2$ and $\Delta CO_2$ are the integrated peak concentrations of SO$_2$ and CO$_2$ when the background concentrations are subtracted (i.e., $\Delta C = C_{peak} - C_{bg}$). By assuming that all sulfur in the fuel has been oxidized to SO$_2$, the FSC can be calculated according to Eq. (10) (Pirjola et al. 2014):

$$
\begin{aligned}
\text{FSC (\%)} &= \frac{\frac{\Delta SO_2(\text{ppb})}{10^3} \cdot M_S}{\Delta CO_2(\text{ppm}) \cdot M_{CO_2}} \cdot EF_{CO_2} \cdot 100 \\
&= \frac{\Delta SO_2(\text{ppb})}{\Delta CO_2(\text{ppm})} \cdot 0.23,
\end{aligned}
\tag{10}
$$

where $M_S$ and $M_{CO_2}$ are the mole masses of S and CO$_2$, and EF$_{CO_2}$ is the emission factor for CO$_2$. Here, the value of 3.107 kg CO$_2$ per kilogram of fuel burned was used (Petzold et al., 2008).

Equation (10) yields lower limits for the FSC (Williams et al., 2009), as a small part of the sulfur in the fuel, less than 6 % (Alföldy et al., 2013) or 0.7 % (Moldanová et al., 2013), might be emitted as SO$_3$ or converted to H$_2$SO$_4$ by homogeneous and heterogeneous pathways in the atmosphere.

## 2.4 Uncertainty estimates

The uncertainty in the measurements is one of the most important issues to solve when analyzing the results. The measurements are influenced by a number of error sources that need to be identified and quantified, including the performance characteristics of gas and particle analyzers and the different probes and sensors used for the measurements. Besides the uncertainties associated with the instruments, the EC and GR methods are very sensitive to the topography and the atmospheric conditions. In particular, the stochastic nature of turbulence (Lenschow et al., 1994; Rannik et al., 2006) and the noise present in the measured signals cause random errors (Lenschow and Kristensen, 1985; Rannik et al., 2016), which are difficult to estimate. However, a number of error sources have been identified that have an influence on the results of flux measurements from the EC method (Businger, 1986; Rinne et al., 2000). The statistical error of an EC estimate is usually quite large.

The uncertainty sources that contribute to the uncertainty of the flux results by the GR method are systematic and random in nature. Calibration of the response of all instruments, correction of the humidity for CO$_2$ analyzers, and cross-calibration due to the sampling tubes of the analyzers are systematic errors. All of the uncertainty sources (systematic and random) that contribute to the results need to be corrected. Even after correction for all systematic errors, there is always a residual component that needs to be included in the uncertainty budget. The residuals are estimated according to the best knowledge available. The variance in the standard uncertainty can be expressed in the following form (JCGM, 2008):

*Atmos. Chem. Phys., 21, 1–20, 2021*        **https://doi.org/10.5194/acp-21-1-2021**

$$u_c^2 = \sum_{i=1}^{n} \left(\frac{\delta f}{\delta w_i}\right)^2 u_i^2 + 2 \sum_{i=1}^{n-1} \sum_{j=i+1}^{n} \frac{\delta f}{\delta w_i} \frac{\delta f}{\delta w_j} u_i u_j \rho_{ij}, \quad (11)$$

where the square root of $u_c^2$ is the combined standard uncertainty. It includes all of the uncertainty components $u_i$ (standard uncertainties) of the function $f$ describing the measurement quantity in question for each of the parameters $w_i$ associated with the results of the measurements. The covariance term (the second term on the right-hand side) in Eq. (11) needs to be taken into account when it is about the same order of magnitude as the independent part in Eq. (11). Once the combined standard uncertainty has been determined, the expanded uncertainty $U$ can be calculated according to $U = \kappa \cdot u_c$. Here, a factor of $\kappa = 2$ was used to represent the 95 % confidence level of a normal distribution. In the case of the gradient method, the uncertainty sources contributing to the fluxes associated with the method can be expressed by applying Eq. (2) to (11). For simplicity, the covariance terms in Eq. (11) have been ignored, as they are of the second order of magnitude.

The combined standard uncertainty of the flux of a compound $c$ can then be approximated in the following form:

$$\frac{u_c(F_c)^2}{F_c^2} = \frac{u(u_*)^2}{u_*^2} + \frac{u(\Psi_h)^2}{\left(\ln\left(\frac{z_2}{z_1}\right) - \Delta\Psi_h\right)^2} + \frac{u(\Delta c)^2}{\Delta c^2}$$
$$+ \frac{u(z)^2}{\left(\ln\left(\frac{z_2}{z_1}\right) - \Delta\Psi_h\right)^2}\left(\frac{1}{z_1^2} + \frac{1}{z_2^2}\right) + u(ws)^2. \quad (12)$$

The standard uncertainties are calculated according to Eq. (11) for each of the contributors in Eq. (12): the friction velocity, the integral functions of the Businger–Dyer functions, the concentrations of gases and particles, and the measurement heights and the wind speed (ws). The estimated relative uncertainties for each of the contributing sources of uncertainty are presented in Table S1 in the Supplement.

The combined standard uncertainty of the fuel sulfur content $u_c$ (FSC) is estimated by applying Eq. (10) to (11), and it can be expressed in the following form:

$$\left(\frac{u_c(\text{FSC})}{\text{FSC}}\right)^2 = \left(\frac{u(\text{SO}_2)}{\Delta\text{SO}_2}\right)^2 + \left(\frac{u(\text{SO}_{2,\text{bg}})}{\Delta\text{SO}_{2,\text{bg}}}\right)^2$$
$$+ \left(\frac{u(\text{CO}_2)}{\Delta\text{CO}_2^2}\right)^2 + \left(\frac{u(\text{CO}_{2,\text{bg}})}{\Delta\text{CO}_{2,\text{bg}}^2}\right)^2, \quad (13)$$

where $u(\text{SO}_2)$ and $u(\text{CO}_2)$ are the standard uncertainties in the measured $\text{SO}_2$ and $\text{CO}_2$ concentrations at the peak area concentration, respectively, and $u(\text{SO}_{2,\text{bg}})$ and $u(\text{CO}_{2,\text{bg}})$ are the respective standard uncertainties in the background concentrations.

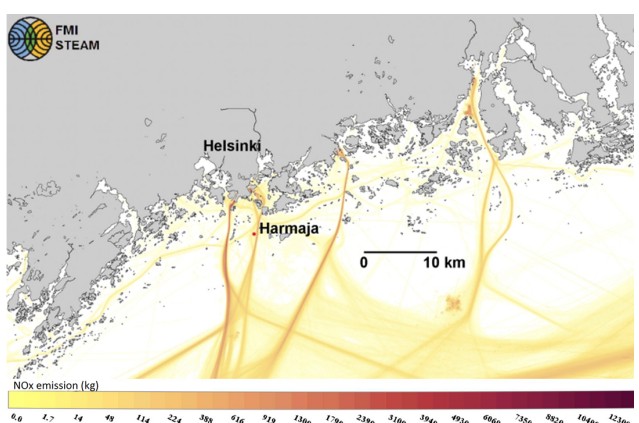

**Figure 1.** International ship routes in the Baltic Sea, the ship routes to the harbors of Helsinki, and modeled $NO_x$ emissions. Land area is marked using gray, sea is shown in white, and $NO_x$ emissions (in kg) are shown using color, as indicated by the color bar. Harmaja Island is marked using a red circle.

## 3 Measurements and data analysis

### 3.1 The measurement site and the meteorological parameters

The first measurement campaign at Harmaja ($60°06'18.166''$ N, $24°58'28.808''$ E) started on 13 July 2011 after installation of the measurement instruments and ended on 12 October 2011. The second campaign started on 7 July 2012 and ended on 20 August 2012. The isle of Harmaja is a pilot station located in the Gulf of Finland, about 4 km from the city of Helsinki in Finland. The ship routes from Tallinn to Helsinki and from Stockholm to Helsinki both pass by the isle of Harmaja at distances of 1 km and 100–200 m, respectively, as shown in Fig. 1. Figure 1 also shows the modeled $NO_x$ emissions.

The measurement station (Fig. 2) was set up in an old military fire control tower made of steel and concrete. All of the measurement instruments were installed inside the tower, and measurement probes and sampling inlets were installed at different heights on a mast beside the tower. The height of the mast was 9 m, and it was located on a breakwater 3.5 m above the mean sea level (a.m.s.l.). The measurement probes were installed at different heights to get an extensive view of the meteorological parameters of interest. Cup anemometers (WAA15 cup anemometers, Vaisala, Finland) measured the wind speed, while the turbulence parameters were measured using an ultrasonic anemometer (ultrasonic wind sensor, uSonic-3, METEK GmbH, Germany). Pt100 sensors measured the ambient temperature. The cup anemometers were installed at three different heights, 12.2, 10.9, and 9.9 m a.m.s.l. The sonic anemometer was installed at the top of the mast at a height of 12.9 m, and the temperature probes were installed at heights of 12.3, 11.0, and

10.0 m. The sampling intakes for the gaseous compounds were installed at two different heights, 12.58 and 9.98 m. The inlets for particle measurements were installed at 8.0 and 10.0 m. The official weather mast of the Finnish Meteorological Institute (FMI) was located next to this mast (Fig. 2) and was equipped with a cup anemometer (WAA15 wind vane, Vaisala, Finland) and a wind direction vane (WAV15, Vaisala, Finland) at a height of 16.6 m. The current sea level was measured as a 30 min average with reference to the mean sea level. The measurement heights of the probes used in the calculations are from the current sea level.

## 3.2   Instrumentation

The atmospheric concentrations of ozone, oxides of nitrogen, and sulfur dioxide were measured simultaneously by conventional gas analyzers intended for ambient air quality measurement. Two identical analyzers of each gas were used to detect the concentration at the two measurement heights (12.58 and 9.98 m). The sampling tubes at each altitude were made equal in length, and PTFE (polytetrafluoroethylene) was used as the tube material, as it is an inert material for each of the gaseous pollutants. The measurement technique used for ozone was the UV-photometric method (EN 14625:2012), and ozone measurements were performed with APOA-360 analyzers (HORIBA, Japan). For nitrogen oxides ($NO_x$), the chemiluminescence method was used (EN-14211:2012), and the measurements were performed with an AC31M analyzer (Environnement S.A., France). The AC31M analyzer was constructed as a two-channel instrument that measures the concentration of NO and $NO_2$ simultaneously. For sulfur dioxide, the UV-fluorescence method (EN-14212:2012) was used, and measurements were carried out with a TEI 43 CTL analyzer (Thermo Fisher Scientific, USA). A LI-7000 instrument (LI-COR, USA) was used for the EC method to measure the concentration of $CO_2$ and $H_2O$, and a Picarro G2301 (Picarro Inc, USA) measured the concentration of $CO_2$, $CH_4$, and $H_2O$. Results of the LI-7000 and the Picarro G2301 were also used to calculate the $CO_2$ flux with the GR method.

For particle sampling stainless-steel tubes with an outer diameter of 12 mm were used. Particle number concentration and size distribution were measured by two ELPIs (electrical low-pressure impactors, Dekati Ltd., Finland) (Keskinen et al., 1992); ELPI1 measured at a height of 10.0 m, and ELPI2 measured at 8.0 m. The measurement principle of both ELPIs was the same: particles were first charged and then classified into 12 stages according to their aerodynamic diameter, in the size range from 7 nm to 10 μm. Both ELPIs were equipped with a filter stage (Marjamäki et al., 2002), and ELPI1 also had an extra stage designed to enhance the particle size resolution for nanoparticles (Yli-Ojanperä et al., 2010). The cutoff diameters were 0.016 (additional stage, only in ELPI1), 0.030, 0.056, 0.093, 0.156, 0.264, 0.385, 0.617, 0.954, 1.610, 2.410, 4.04 (only in ELPI2), and 9.97 μm. The mass concentration of particles smaller than 1 μm ($PM_1$) was calculated by assuming the particles to be spheres with a density of 1000 kg m$^{-3}$.

The strategy for the air quality measurements in 2012 was different: it included the measurement of concentrations of NO, $SO_2$, $CO_2$, $O_3$, and particles at one height, and $CO_2$ fluxes using the EC method. The $NO_x$ analyzer was a modified version of a single-channel analyzer (Thermo 42 CTL by Thermo Fisher Scientific, USA). The modification was made by bypassing the $NO_2$ converter and the solenoid valve, increasing the flow rate by choosing an orifice which allowed a flow rate of up to 2 L min$^{-1}$, and by using the shortest integration time. The purpose of the modification was to make the analyzer faster for detecting the emission peaks of the ships, although for NO gas only. The $SO_2$ analyzer was also a modified version of the TEI 43 CTL instrument (Thermo Fisher Scientific, USA). The purpose of the modification was the same as in the case of the $NO_x$ analyzer: to make it faster for detecting ship emissions. The $O_3$ and $CO_2$ analyzers and the particle analyzer (ELPI1) were the same as in the 2011 campaign.

## 3.3   Calibration of the instruments

The temperature probes (Pt100) were calibrated in the calibration laboratory at the FMI for meteorological quantities. The wind speed anemometers were serviced (cleaned and the ball bearings changed) in the same laboratory. The gas analyzers were calibrated in the reference calibration laboratory at the FMI before and after the field campaign. The calibration laboratory is responsible for the tasks of the national reference laboratory on air quality, and it conducts the calibrations of the air quality analyzers and calibration facilities in Finland. The laboratory maintains the traceability of the calibration to the SI units, and it is accredited according to ISO 17025:2005 for all measured gas compounds except $CO_2$. The dynamic dilution method by accurate gas sources was used for the calibration (Haerri et al., 2017). The calibration concentrations were selected to cover the expected measurement ranges for each of the gas components.

The analyzers were also calibrated at the measurement site during the campaign using a field calibration unit similar to that in the laboratory. Both ELPIs were factory calibrated and serviced. Zero setting and high-efficiency particulate absorbing filter (HEPA) filtration tests were performed before and after each measurement period. Based on the parallel measurements of the ELPIs on 30 August and 2 September 2011 correction factors were inferred for ELPI2, separately for each stage ($dN/d\log Dp$), and for the number concentration of particles smaller than 1 μm ($N_{tot}$) that was used for the flux calculations (more details are given in the Supplement). All measured $N_{tot}$ data from the ELPI2 were corrected accordingly. This ensures that the results are correct within the stated uncertainty and are comparable with the other similar measurements.

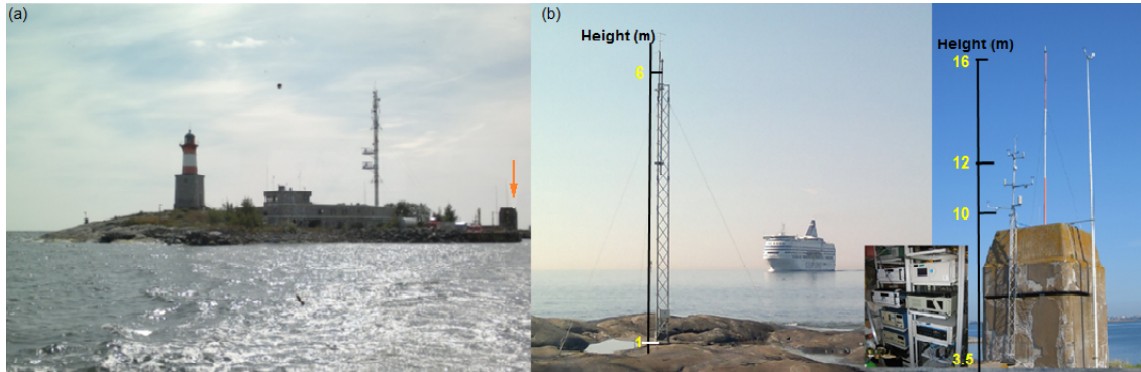

**Figure 2. (a)** A photograph of Harmaja Island; the orange arrow points to the measurement tower. **(b)** Measurement instrumentation was installed in racks inside the fire control tower. The mast on the right includes all of the measurement probes (wind and temperature probes and sample intakes) for the gradient and eddy-covariance methods during the 2011 campaign, whereas the mast on the left includes the probes and sample inputs for the eddy-covariance method and for measuring gas and particle concentration during the 2012 campaign. Both masts faced the open sea. The masts stood approximately 20 m from the shoreline and 3.5 m a.m.s.l. in 2011, and 3 m from the shoreline and 1 m a.m.s.l. in 2012. The official weather mast of the FMI is located at the side of the tower (right-hand mast, 16.6 m).

## 3.4 Data acquisition systems and data analysis

The data acquisition systems consisted of several components. The meteorological measurements were collected and stored by a MILOS 500 system (Vaisala, Finland). The ambient air quality gas analyzers were connected to a Envi-Das 2000 (Envimetria, Israel) data collection system, and the sonic anemometer and the LI-7000 were connected to a fast data acquisition system; the Picarro G2301 instrument used the system provided by the manufacturer, and the ELPI software was used for the collection of particle data.

The times, East European Time (EET+2) at Coordinated Universal Time (UTC)CE3, of the different data acquisition systems were synchronized during each calibration and maintenance event. The final adjustment of all data sets at each of the altitudes was made manually from the time series so that the obvious peaks coincided. Data collection for the EC method was performed at 10 Hz time resolution. In the case of the GR method, data were collected as 15 s and 1 min averages, whereas the total number concentration of particles in the size range of 7–1000 nm ($N_{tot}$) and $PM_1$ were collected at 10 s intervals. The meteorological data from the official weather mast were collected as 10 min averages. A consistent data set was formed as 30 min averages from the synchronized individual data acquisition systems.

The first target for the data analysis was to achieve accurate and good-quality continuous time series for the gaseous compounds and particles at each of the measuring heights. Secondly, the turbulence parameters (M–O length, stability parameter, friction velocity) were needed for calculating the transfer coefficients $K_m$ and $K_c$, the dimensionless gradient, and their integral functions for heat and momentum, and for calculating the fluxes of the chemical compounds and fine particles. The friction velocity and the M–O length, which were calculated from the data obtained by the sonic anemometer, were used as input parameters for the GR method. The benefit of this was to reach better agreement with the flux parameters.

## 3.5 Measurements on R/V *Aranda*

In 2012, measurements were also made on the Finnish research vessel *Aranda* during 2 d. The ship was kept stationary at a point approximately 2 km SSW of the measuring mast at Harmaja, with no islands between Harmaja and the ship. The bow of the ship was equipped with a sonic (METEK USA-1, Germany), an open-path LI-7500, and an enclosed-path LI-7200 at two heights (10 and 16 m) for the measurement of $CO_2$, $H_2O$, and momentum and heat fluxes. During the measurements, the ship's bow was kept within $\pm20°$ of the oncoming wind direction. The sonic measurements were corrected for ship motions with a motion sensor, MRU6 (Kongsberg, Norway), according to Drennan et al. (1994). In this study, the measurements from the sonic and enclosed-path LI-7200 at the height of 16 m were used, and the calculated fluxes were corrected for water vapor fluctuations according to Webb et al. (1980). The partial pressure of $CO_2$ in the surface water at a depth of 4 m obtained from the ship's flow-through system was measured continuously with an equilibrator and a LI-6262. The measurements were transformed to in situ water temperature according to Takahashi et al. (1993). All of the LI-COR instruments were calibrated against 0, 364, and 700 ppm $CO_2$ gases.

## 4 Results

### 4.1 General overview

Environmental factors (e.g., the fire control tower) caused challenges with respect to the measurement signals. Al-

though the measurement probes were installed at different heights above the top of the tower, the measurement signals were affected by disturbances in the flow field. Therefore, a computational fluid dynamic (CFD) program Open-FOAM (version 7; https://www.openfoam.org, last access 7 April 2020) was used to determine the amount of distortion and the required correction. OpenFOAM is a C++-based open-source software developed mostly, but not exclusively, for CFD. The airflow around the shoreline and the measuring structure was modeled using steady, incompressible, single-phase potential flow. The simulation covered a 80 m long, 40 m wide, 30 m high rectangular box around the measurement area.

Figure 3a illustrates the calculated wind field isopleths at a wind speed of 9 m s$^{-1}$ over the open-sea area, and it shows how the flow field is disturbed around the measurement mast. Based on the calculated isopleths, we determined the corresponding height over the open sea for each measurement height. The actual heights of wind speed probes are shown at the measurement mast on the right of Fig. 3a, and their projected heights over open sea are shown on the left. The heights at the measurement masts for wind speed measurements were reduced from 16.63, 12.88, 12.18, 10.88, and 9.88 to 15.5, 11.1, 10.2, 8.6, and 7.2 m, respectively. Similarly, the heights of the sample intakes were reduced from 12.88, 12.58, 10, 9.88, and 8 down to 11.1, 10.7, 7.38, 7.2, and 4.7 m, respectively. The correction for the wind speed was calculated from different simulations varying the wind speed and comparing the calculated results at the measurement mast with the open-sea area. These calculations show that a linear relationship for the wind distortion at the measurement mast compared with the open-sea area was good enough to correct the observed wind speed measurement at all heights. The recalculated wind speed profiles (six profiles, cases 1 to 6) at each of the corrected measurement heights are presented as averages over short periods (2 to 3 h) in the wind sector to the open sea (i.e., $150° \leq \text{wd} \leq 270°$, where wd CE4 denotes wind direction). The data were roughly classified into three categories: code 1 – the M–O theory is valid (no swell, $c_p < \text{ws}$, where $c_p$ and ws are the peak wave and wind speeds, respectively); code 2 – the M–O theory is possibly valid (moderate swell, $\text{ws} < c_p < 2 \cdot \text{ws}$); and code 3 – the M–O theory is not valid (dominant swell, $c_p > 2 \cdot \text{ws}$). In Fig. 3b, the wind speed profiles (cases 1 to 6) are presented in situations where the M–O theory is valid (code 1) and where the M–O theory is not valid (code 3). In Fig. 3c, the ambient air temperature profiles are calculated from the same situations as the wind profiles. In Fig. 3d, the wind rose at the height of 15.5 m shows the patterns of prevailing wind sectors with wind speed ranges.

As an example, Fig. 4a–b depicts the time series of 1 min averaged concentrations of the measured gas compounds at 10 m altitude during the first campaign. The sharp peaks in the concentrations of nitrogen monoxide are very striking. In a detailed examination, the duration of the emission peaks

from the ships were of the order of a few minutes. Where there was a peak in the NO concentration, a negative peak was also detected in the ozone concentration due to the fast reaction producing $NO_2$. This was observed within seconds following the emission from the ship into the atmosphere. The changes in the concentrations of NO, $NO_2$, and $O_3$ were equal (i.e., the change in the concentration of NO took place according to the stoichiometric balance).

The concentration of sulfur dioxide was very low, but clearly distinguishable peaks, with a maximum of 28 ppb, were observed in the data. These peaks originated from the passing ships. The ship peaks were also seen in the $CO_2$ data, but the short-term variation in the $CO_2$ concentration might be larger than the contribution from the ship emissions (Fig. 4b). When looking at the particulate concentrations $N_{\text{tot}}$ and $PM_1$ (Fig. 4d), interesting features can be observed. Ship peaks can be clearly distinguished; however, the background levels of $N_{\text{tot}}$ and $PM_1$ were highest on 28 August 2011 in the morning when the wind blew from the south. The 96 h backward-trajectory analysis of FLEXTRA by NILU (Norwegian Institute for Air Research; Stohl et al., 1995) showed that, in the measurement period before noon on 28 August, an air mass was transported through central Europe and arrived in Helsinki (Fig. 5a) carrying anthropogenic pollutants. A sudden drop in the concentrations of $N_{\text{tot}}$ and $PM_1$ occurred at noon on 28 August when the wind turned and blew from the west until 10:00 UTC on 30 August (Fig. 4c) over the Atlantic and Baltic Sea carrying clean air with low particulate concentrations (Fig. 5b). Simultaneously, the diurnal variation in $CO_2$ diminished. During that period there was no precipitation. During the last 12 h before the clean air mass arrived in Harmaja, the average background particle concentrations stayed rather constant at $\sim 2.7 \times 10^3$ cm$^{-3}$, whereas the $PM_1$ increased from $\sim 4$ to $\sim 11$ µg m$^{-3}$. This indicates that larger particles were also transported from Europe. In fact, this is obvious from Fig. 6, which presents the average number size distribution (Fig. 6a) as well as the volume size distribution (Fig. 6b) of background particles in the evening of 27 August, at noon on 28 August just before the clean air mass arrived, in the afternoon of 28 August, and early in the morning of 1 September. In the latter case, the particle number concentration was highest ($3.6 \times 10^3$ cm$^{-3}$), but due to the small particle sizes (Fig. 6a), they did not have an effect on the volume (and mass) size distribution (Fig. 6b).

## 4.2 Concentration roses

Figure 7 presents the maximum values for the concentrations of each gas compound and the particle number within 10° wind sectors at both measurement heights. These results are in good agreement with those based on the sum of the measured concentrations, showing the cumulative contribution from different wind sectors at the measurement point (not shown). For $NO_x$ and $SO_2$, similar patterns were observed, indicating the ship routes in sectors 90 to 120, 150 to 180, and

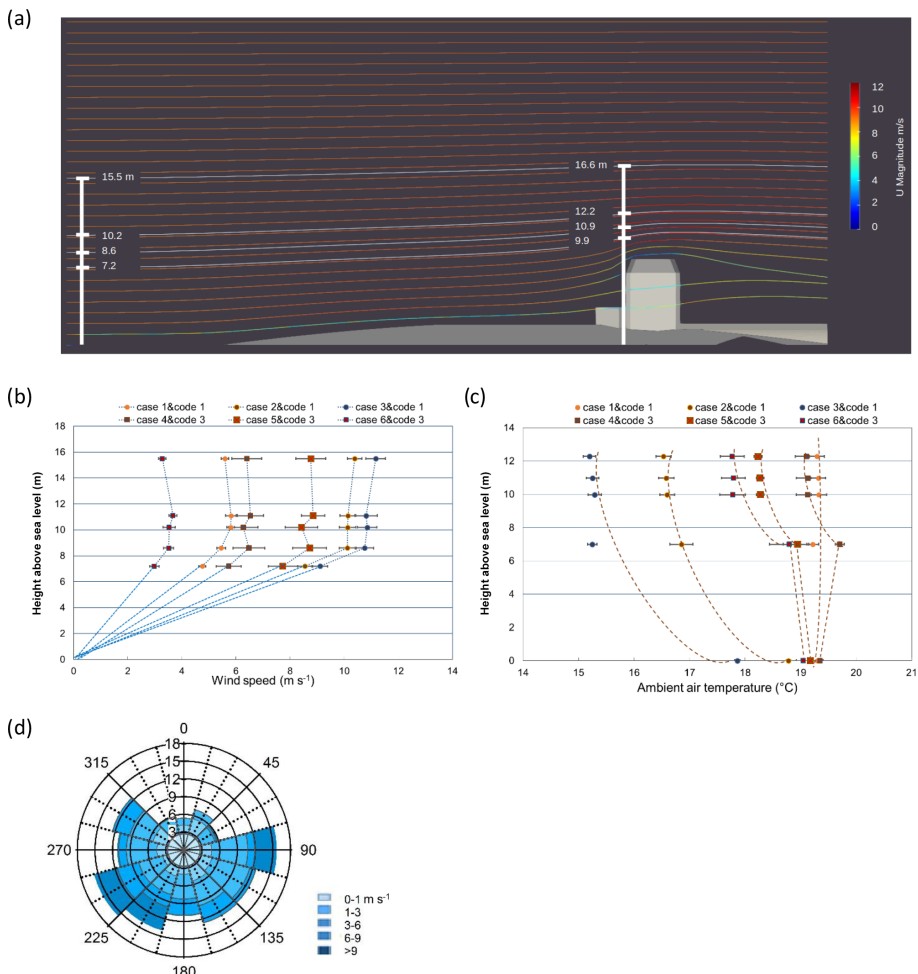

**Figure 3. (a)** The actual measurement heights were reduced to the corresponding heights over the open-sea surface utilizing calculated isopleths from a flow dynamics program. The actual wind speed probe heights at the measurement mast are shown on the right, and their projected heights over the open sea are shown on the left. **(b)** Mean wind speed profiles over 2 to 3 h are presented at different heights in the wind direction sector from 150 to 270°. The dots represent the situations with no swell (code 1), whereas the squares represent cases with swell (code 3). The error bars show the standard deviation of the wind speed over the average period at the respective heights. **(c)** Hourly mean temperature profiles are presented for the same situations as the wind profiles shown in Fig. 3b. The error bars show the standard deviation of the temperature over the average period at the respective heights. **(d)** Wind rose of direction (scale as a percentage) and wind speed (m s$^{-1}$) at different ranges at the measurement height of 16.6 m.

270 to 300°. In addition, there is a clear difference in concentrations between the heights: the higher concentrations were mostly at the highest measurement level. In the case of oxygen compounds ($= O_3$ and $O_3 + NO_2$) and $CH_4$, the patterns were more evenly distributed. In the case of $CO_2$ and $N_{tot}$, there is an indication of the ship routes but also of the influence of the city of Helsinki.

## 4.3 Fluxes

Quality control (QC) and quality assurance (QA) procedures are actions that should be considered in order to improve data quality and make data comparable with similar data from other studies. Although QA and QC procedures have slightly different meanings, in this study, the quality assurance and quality control (QA/QC) procedures are considered together. The following QA/QC procedures and criteria for flux calculations were taken into account:

1. calibration of the analyzers used for gases and particles (Sect. 3.3) – for the GR and EC methods;

2. criteria for the minimum concentration difference between the measurement heights (Fig. S2 in the Supplement) – for the GR method;

3. correction of the wind flow field around the measurement mast according to the CFD calculations (Sect. 4.1) – for the GR method;

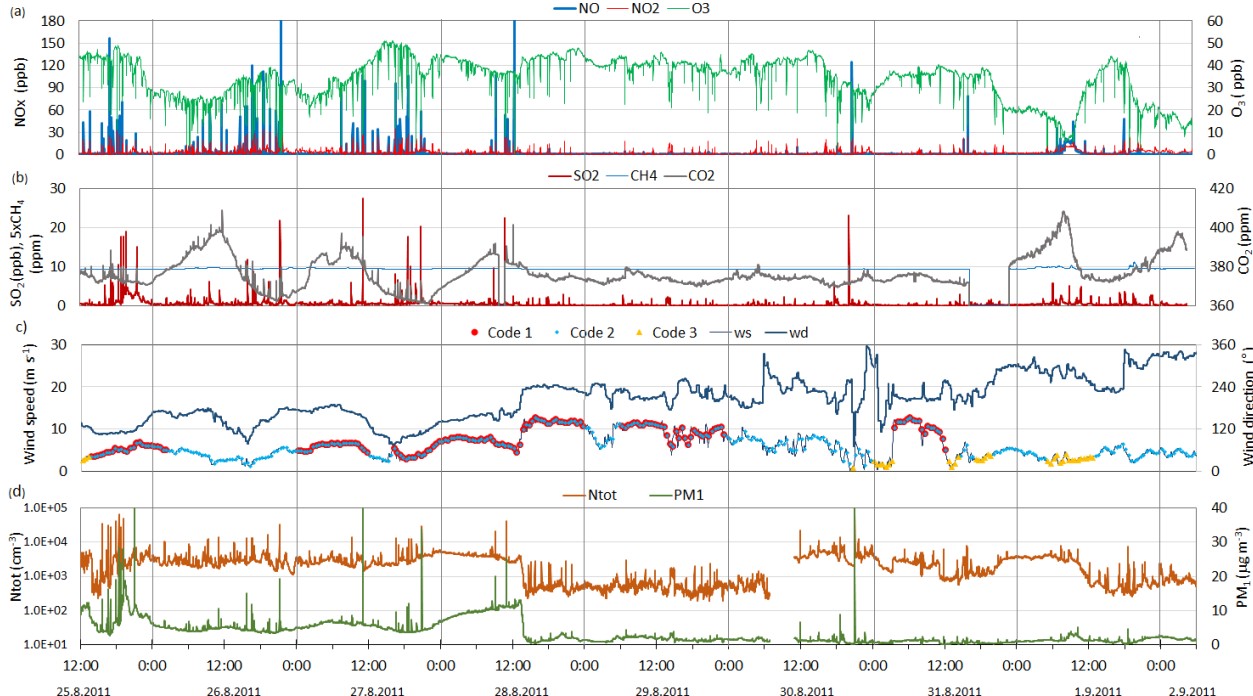

**Figure 4.** Time series of 1 min average concentrations of gaseous **(a–b)** and particulate matter **(d)** as well as the wind speed and wind direction **(c)** during the period from 25 August to 2 September 2011. Also shown in panel **(c)** is the 30 min average wind speed in different situations with respect to M–O theory (i.e., codes 1 to 3).

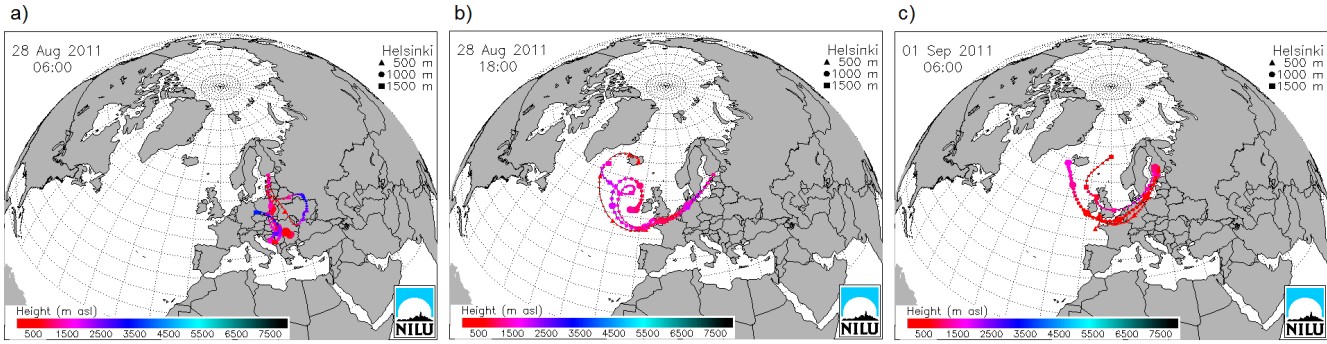

**Figure 5.** Selected air mass trajectories on 28 August at 06:00 UTC **(a)** and 18:00 UTC **(b)** and on 1 September at 06:00 UTC **(c)** during our campaign in 2011. Please note that the times are given in UTC, whereas our data sets are given in EET (UTC+2).

4. restriction to open sea, i.e., wind direction in the range of 150–270° (Fig. 3a) – for the GR and EC methods;

5. analysis of swell to determine the validity of M–O theory with codes 1–3 (Sect. 4.1) – for the GR method;

6. the footprint area was estimated at each of the measurement height under at neutral, stable, and nonstable conditions (Fig. 8) – for the GR and EC methods;

7. stationarity criteria following the criteria of Foken and Wichura (Foken and Wichura, 1996) – for the for GR and EC methods;

8. the intermittency was applied according to Mahrt et al. (1998) – for the EC method;

9. WPL correction due to water vapor and heat flux – for the GR and EC methods;

10. cross-sensitivity of the compounds on the used analyzers – for the GR and EC methods;

11. preparation of the uncertainty budget for the measurement results – for the GR and EC methods.

The footprint area (i.e., the area upwind where the exchange of gases and particles between the air–sea surface are

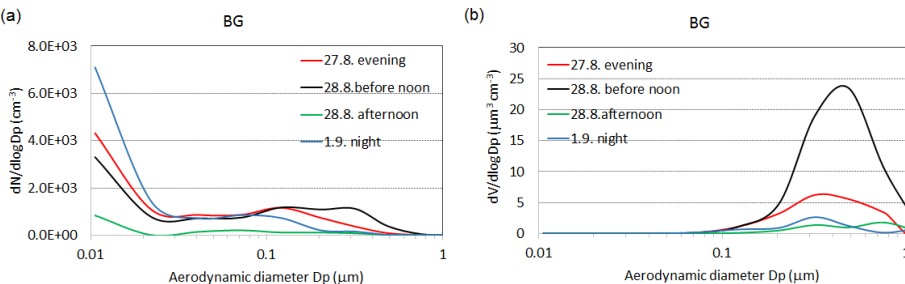

**Figure 6.** Number **(a)** and volume **(b)** size distributions of background particles.

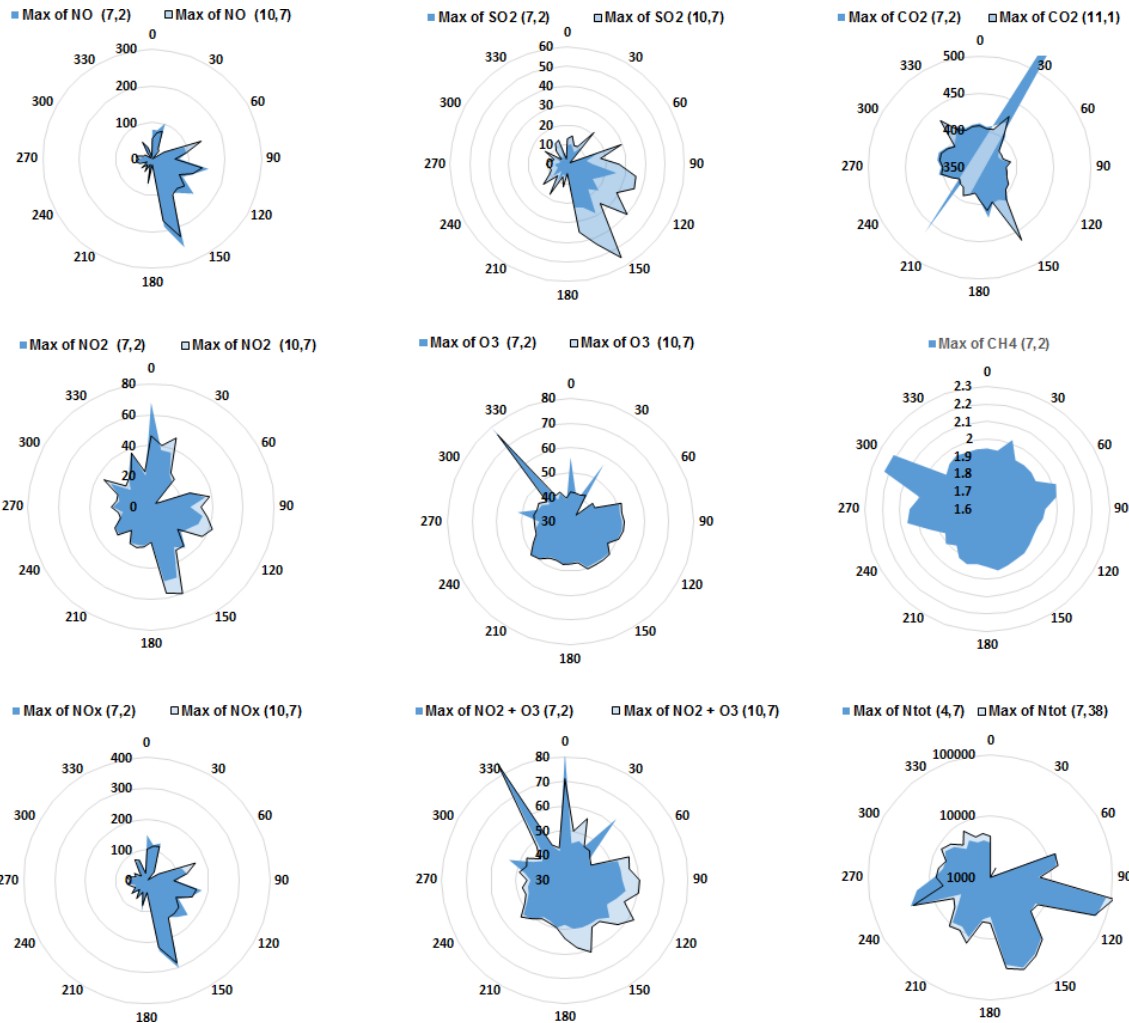

**Figure 7.** Roses of the maximum gaseous concentrations of nitrogen oxides (NO, NO$_2$, NO$_x$; in ppb), oxidants O$_3$ and NO$_2$+O$_3$ (in ppb), SO$_2$ (in ppb), and CH$_4$ and CO$_2$ (in ppm). Also shown are the roses of particle concentrations $N_{tot}$ (in cm$^{-3}$). All roses except for CH$_4$ were plotted for the two heights; the corrected heights (m) are given in parentheses above the roses. The direction of the coastline is from 240 to 60°.

expected to be a source of the measurement results) was calculated according to Högström et al. (2008). The footprint area was calculated at each of the measured heights under stable, neutral, and unstable conditions. Figure 8a illustrates the relative intensity of the footprint area under neutral conditions as a function of upwind distance from the measurement mast at instrument heights of 4.7, 7.2, and 10.7 m. Stable and unstable conditions are presented in Fig. S3. The cumulative relative contribution (Fig. 8b) indicates that less than 0.3 % of the observed flux at the lowest height (4.7 m) takes place at

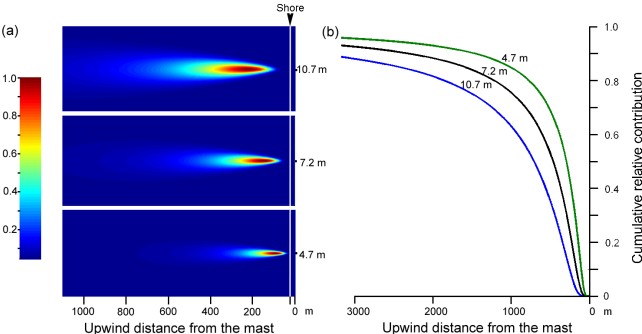

**Figure 8.** **(a)** Flux footprint areas at neutral stability seen by the $CO_2$ instruments at altitudes of 10.7 and 7.2 m and by the ELPIs at altitudes of 7.2 and 4.7 m, as recalculated in Fig. 3a. The $x$ axis refers to the upwind distance from the instruments, and the color bar shows the relative intensity of the sea surface area to the flux. Panel **(b)** shows the cumulative relative contribution.

a distance of 20 m from the mast, reaching 90 % at a distance of 3 km. At a height of 10.7 m, the footprint area starts at 40 m from the mast and reaches 85 % at distance of 3 km. The storage fluxes were not considered in this campaign. The site was by the sea where the turbulent mixing was most likely the main driving force for gas and particle dispersion most of the time.

The stationary requirement was calculated using the method proposed by Foken and Wichura (1996) for fluxes of $CO_2$, momentum, water vapor, and heat. Similarly, the effect of flux intermittency was estimated using the index proposed by Mahrt et al. (1998).

Cross-sensitivity of the compounds (e.g., water vapor) on the response of the analyzers used are included into the uncertainty budget or corrected directly in the results (see Fig. S1). The influence of NO and $NO_2$ compounds on the response of the $SO_2$ analyzer was tested in the laboratory. Known concentrations of NO and $NO_2$ gases were injected into the $SO_2$ analyzer to define the response function for both NO and $NO_2$. The results of the $SO_2$ analyzer were then corrected accordingly.

The uncertainty sources of the measurement results for fluxes using the gradient method are presented in more detail in Table S1. To estimate the uncertainty of the momentum flux and $CO_2$ flux measurements using the EC method, we calculated the expected statistical variability using the cospectrum. The statistical variability of the covariance (which equals the integral of the cospectrum) depends on the shape of the cospectrum. A white cospectrum implies lower statistical variability for the covariance than a peaked cospectrum. In our estimates, we have taken the observed shape of the $CO_2$ cospectrum into account when calculating the estimate of the statistical variability of the covariance (see, e.g., Bendat and Piersol, 2010, Chap. 7). For the momentum flux it was 20 %, and for the $CO_2$ flux it was 30 %. This wide uncertainty range is typical in real meteorological

**Table 1.** Estimated relative expanded uncertainties for the fluxes of gaseous $CO_2$ and nanoparticles using the GR method, and for $CO_2$ using the EC method under stationary meteorological conditions ($U_{\text{Stat.Met.}}$). More details on the uncertainty budget are presented in Table S1. The uncertainty values are given at the median flux values.

| Flux | Method | Flux (median) | | $U_{\text{Stat.Met.}}$ |
|------|--------|--------------|---|--------------|
| $F_{CO_2}$ | EC | $-0.081$ | $\mu\text{mol m}^{-2}\,\text{s}^{-1}$ | 30.0 % |
| $F_{N_{\text{tot}}}$ | GR | $-229.1$ | $\text{cm}^{-3}\,\text{m s}^{-1}$ | 30.8 % |

situations and explains the scatter in the EC estimates in, e.g., Fig. 12. The analysis of uncertainty follows the guidelines provided by the Joint Committee for Guides in Metrology (JCGM, 2008). Based on the analysis, the relative expanded uncertainties for the flux measurements of $CO_2$ and nanoparticles are presented in Table 1 under stationary meteorological conditions.

The eddy diffusivities $K_h$ and $K_m$ describe the turbulent mixing conditions for momentum and heat. $K_m$ also describes the behavior of the gaseous compounds and particles, as explained in Sect. 2.1. In Fig. 9a, the eddy diffusivities $K_h$ and $K_m$ at the measurement level of the sonic anemometer are presented as a function of stability. From Fig. 9a, b, and d, it is also evident that swell mostly occurs under very unstable conditions in these data and is only observed under stable conditions in a few cases; in contrast, situations with no swell occur under near-neutral conditions. Wind speed and friction velocity in Fig. 9c show a clear dependence on the wind direction. A linear relationship between the average wind speed and the friction velocity is seen in the sectors where the wind arrives over an open-sea area, whereas nonlinear behavior is seen towards the northern sector (345 to 45°), where there are more obstacles. The gradient function for momentum in Fig. 9d has a significant spread around the Businger–Dyer gradient function in the wind sector from 150 to 270°.

Dispersion of a ship plume is schematically presented in Fig. 10a. The black curves denote the edges of the plume. When the lower curve reaches the sea surface, the pollutants would be reflected from the sea surface if there were no pollutant flux to the sea. Near the surface, the sum of the incident and reflected concentrations will add up to the same value if the reflection is total, forming a new boundary layer where the vertical concentration gradient of the pollutants vanishes. If there is a flux to the sea surface, this will result in a vertical gradient of pollutants in this new boundary layer. The fluxes to the sea from the ship can be measured when the plume is over the flux footprint if the measurement instrument is inside the new boundary layer. As an example, Fig. 10b illustrates the momentary plumes at the sea surface for a ship traveling to the city of Helsinki and passing Harmaja Island at a speed of 21.5 kn ($\sim 11\,\text{m s}^{-1}$). The arrows show how the apparent plume is generated in the $(u, v)$-coordinate system

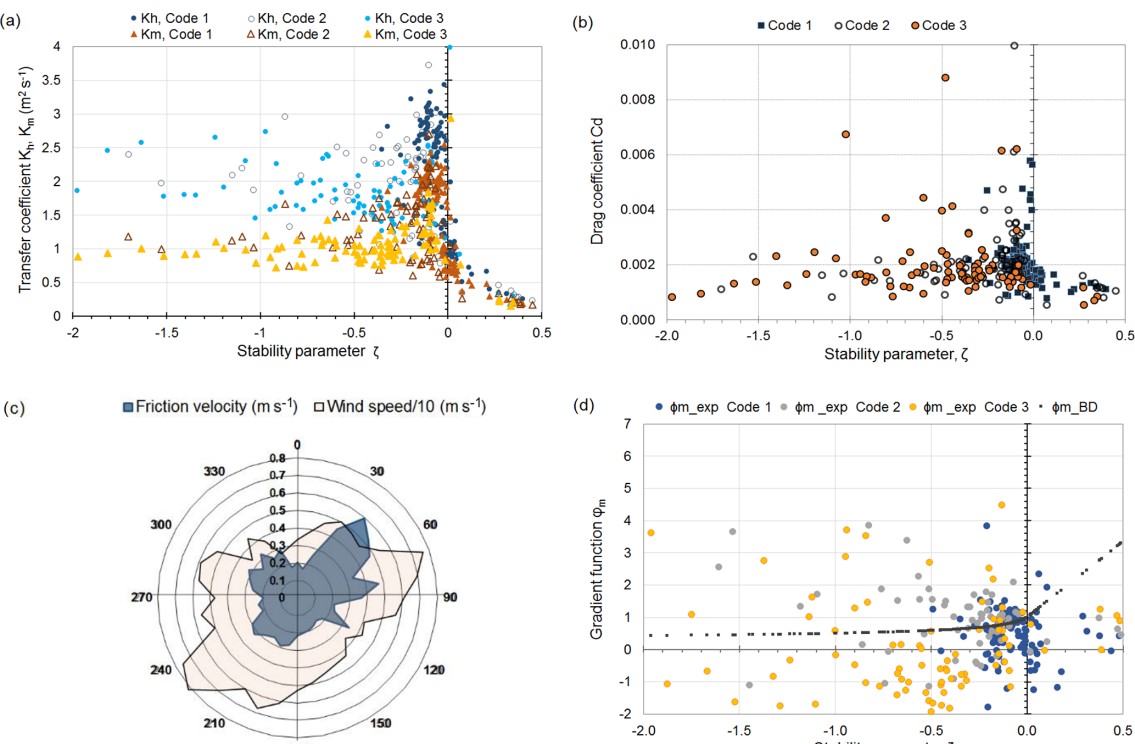

**Figure 9. (a)** The eddy diffusivity coefficients $K_h$ (blue) and $K_m$ (brown) as a function of stability at a height of 12.28 m with wave codes 1, 2, and 3. **(b)** The friction velocity from sonic anemometer measurements as a function of stability at a height of 11.1 m with wave codes 1, 2, and 3. **(c)** The rose of friction velocity and wind speed (divided by 10) as averages over 10° wind sectors. **(d)** The gradient functions from the experiments ($\phi_m\_$exp) at 15.5 m altitude and from the Businger–Dyer formula ($\phi_m\_$BD). In panels **(a)**, **(b)**, and **(d)**, the stationary criteria in the wind sector from 150 to 270° are based on the momentum flux.

as well as where the pollutants advecting to the footprint area come from. In the example, the wind speed was 11 m s$^{-1}$ and the wind direction was 216°, as in the afternoon and evening of 28 August (Fig. 4c). The momentary plume figures are shown 15, 23, 30, and 37 min after the start, and the plume concentration gradients decrease as the plume moves further. At the footprint area, the gradient is really small, indicating a horizontally homogeneous situation. If the stationary criteria for heat, water vapor, and momentum are also valid, the momentary vertical gradients give the momentary flux.

Before the calculation of fluxes of gases and particles using the GR method, it is necessary to evaluate the measured concentration differences between the measurement heights as well as their uncertainty limits. The uncertainty of the gas and particle analyzers as a function of concentration is presented in Fig. S1, and the concentration differences between the measurement heights is shown in Fig. S2. Based on the analysis, the concentration differences for $N_{tot}$ clearly exceeded the uncertainty limit, enabling the calculation of the $N_{tot}$ fluxes using the GR method. In the case of raw $CO_2$, the concentration difference frequently exceeded the uncertainty limit, although only rarely in cases with WPL correction, i.e., in dry air. Consequently, the fluxes for $CO_2$ were too small to be detected using the GR method. On the contrary, the $CO_2$

fluxes obtained using the EC method did not suffer from the same problem (direct flux method), and those fluxes were acceptable except in cases when the stationary criteria were not fulfilled. We also made the comparison of the GR and EC methods with sensible heat (Fig. S4). A clear correlation between the methods can be observed if the calculation of the sensible heat using the GR method is made from the sea surface up the measurement heights of 11 and 15 m. However, the temperature difference between the measurement heights (11 and 15 m) was mostly too small to be detected, and no flux could be calculated.

Figure 11a illustrates the time series for the $CO_2$ flux using the EC method, and Fig. 11b shows the $N_{tot}$ flux using the GR method along with the uncertainties. Only the fluxes that fulfilled the stationary criteria with no swell in the 150–270° wind sector were taken into account. Most of the time, the $CO_2$ flux was positive (upward) in Fig. 11a, whereas the flux of $N_{tot}$ was mostly negative (downward) in Fig. 11b. The WPL correction of the $CO_2$ flux from the EC method corrects wet air to dry air and corrects for the water flux. Due to the damping of temperature fluctuations in the long sample tube, the WPL correction for heat flux was insignificant (Rannik et al., 1997). During the period when the air masses were moving from the Atlantic (from 28 to 31 August), the correction

(a)                                                              (b)

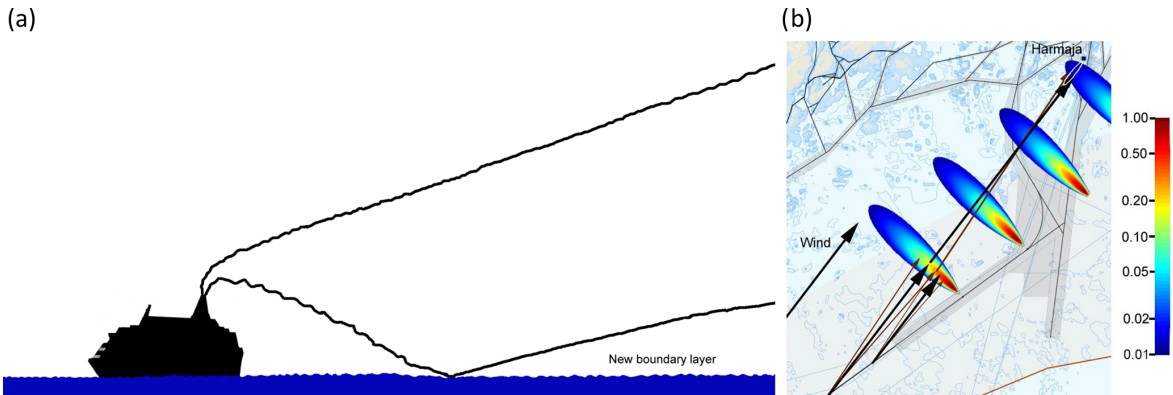

**Figure 10. (a)** Schematic figure of the vertical profile of ship plume dispersion. If the stack height is around 55 m, the plume touches the sea surface at a distance of around 450 m from the ship. Panel **(b)** shows the momentary plumes at 15, 23, 30, and 37 min after start (0 min) at the sea surface. The black arrows are the wind speed (ws) vectors (ws = 11 m s$^{-1}$), the brown arrows show the dispersion of the plume, and the gray lines are the ship routes to Helsinki, with one passing Harmaja Island on its right side. Note that the color bar is shown using a logarithmic scale (2 decades, highest values dark red) and refers to the pollutant concentrations in the chimney. The footprint area is shown using a white ellipsoid at Harmaja, inside of which 80 % of flux is reached at the height of 10.7 m.

due to the water flux dominated and caused a positive offset to the $CO_2$ flux (Fig. S5). The observed $CO_2$ fluxes are also in line with the previous measurements (Honkanen et al., 2018) from Utö Island in the Baltic Sea.

Assuming that there is no inversion below the ships' chimney height, profile measurements of the compounds indicate that the deposition of nanoparticles towards the sea surface is most probably caused by ship emissions (Fig. 11c). More detailed analysis was made to compare the fluxes of $CO_2$ and $N_{tot}$ calculated from single ship plumes, which lasted from 3 to 7 min with the 30 min fluxes. Figure 11c and d show clear peaks in both $CO_2$ and nanoparticles, but the contribution of these ship peak fluxes was insignificant compared with the 30 min fluxes (not shown in the figure).

From the campaign in 2012, the $CO_2$ fluxes from the EC methods measured at Harmaja and on R/V *Aranda* are presented in Fig. 12a. The corrections to the $CO_2$ fluxes measured from humid air samples have been criticized as being insufficient (e.g., Landwehr et al., 2014). The fluxes in this study mainly fulfilled the criteria presented in Miller et al. (2010), Sahlée et al. (2011), Blomquist et al. (2014), and Landwehr et al. (2014). The $CO_2$ fluxes measured on R/V *Aranda* might be underestimated because the flow rate was too small, 10 slpm (standard liters per minute). No systematic differences were observed when the LI-7200 $CO_2$ fluxes were compared with the LI-7500 on the ship mast and also with the measurements made at Harmaja.

The WPL correction performed to both of the data sets in 2011 and 2012 was small in general (see Fig. S5). In 2011, the sign of the flux was changed after WPL correction during the period between 28 and 31 August. Generally, positive $CO_2$ fluxes are common in winter and occasionally in summer after the spring algae bloom, as $CO_2$ is less soluble in the warming seawater. In the Baltic Sea, however, the blue-green

algae bloom extends the biologically active season, and the positive fluxes in coastal regions are mainly caused by frequent upwelling events (e.g., Lehmann and Myrberg, 2008; Norman et al., 2013). The $CO_2$ flux from the EC method and the partial pressure of $CO_2$ in seawater were measured at the same time as the R/V *Aranda* measurements in late July 2012; the large difference between the partial pressures in seawater and in air indicates the upwelling event (Fig. 12c) causing the positive $CO_2$ fluxes in Fig. 12a. The scatterplot of the $CO_2$ fluxes at Harmaja and on R/V *Aranda* is presented in Fig. 12b. The large scatter is most probably due to the different locations: the measurements of $pCO_2$ (Fig. 12c) suggest that the $pCO_2$ was not spatially homogeneous. The estimated statistical variability is large and also contributes to the scatter. The sensible heat fluxes (Fig. 12d) were calculated using the EC method during the 2012 campaign at Harmaja and at two heights (10 and 16 m a.s.l) on R/V *Aranda*, which showed good agreement with each other.

Unfortunately, direct comparison between the $CO_2$ fluxes using the EC and GR methods at Harmaja in 2011 as well as in 2012 on R/V *Aranda* could not be performed due to the fact that the $CO_2$ fluxes were too small to be detected by the GR method. Discrepancies between the measurement results using the GR and EC methods have been reported in the literature (Myklebust et al., 2008; Muller et al., 2009) for $CO_2$ and for $O_3$, but no systematic reason has been found.

## 4.4   Fuel sulfur content

The FSC was determined in the measurement campaigns in both 2011 and 2012. The two campaigns differed from each other with respect to the measurement strategies as well as with respect to the data collecting frequency, once per minute in 2011 and every 15 s in 2012. In both cases, the data acqui-

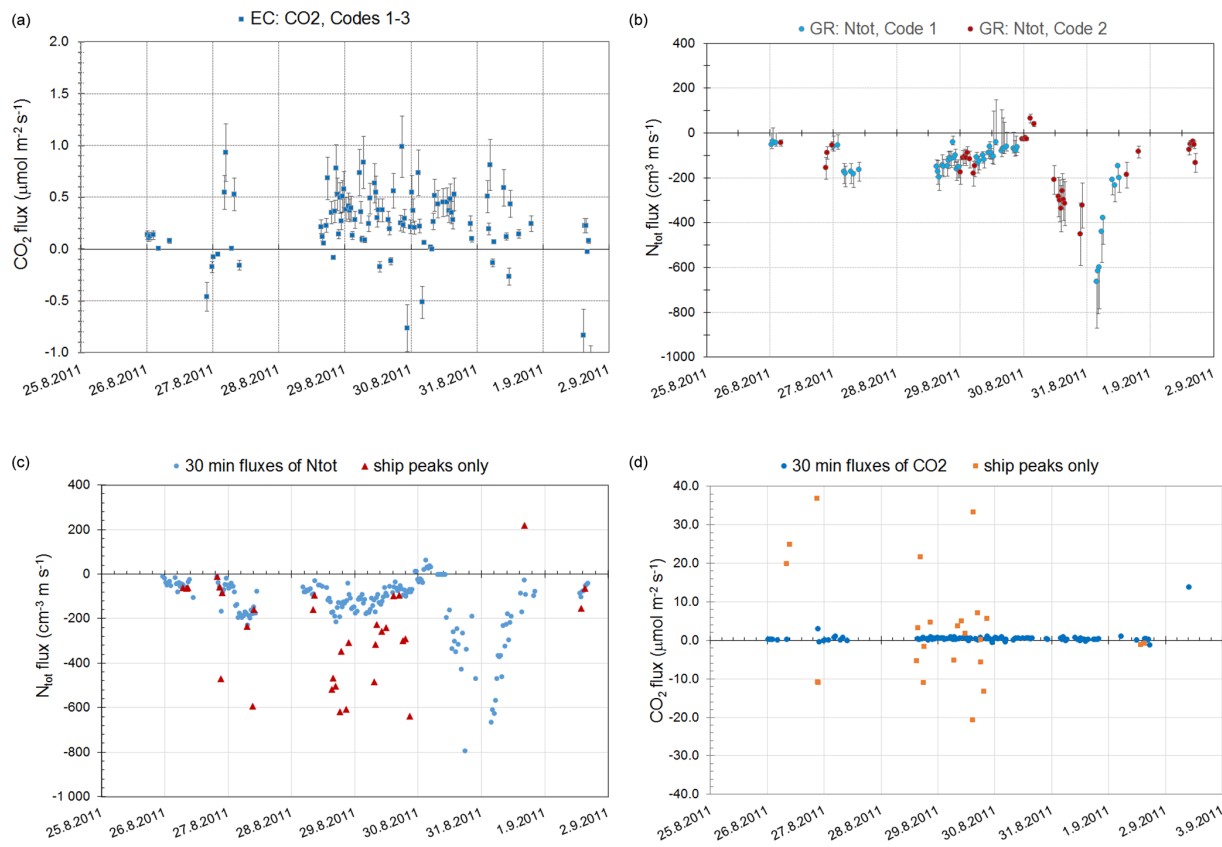

**Figure 11.** Time series of 30 min fluxes for $CO_2$ ($\mu$mol m$^{-2}$ s$^{-1}$) using the EC method **(a)** under all wave conditions (codes 1 to 3), and $N_{tot}$ (cm$^{-3}$ m s$^{-1}$) using the GR method **(b)** with no swell (codes 1–2), along with the uncertainties. The data in panels **(a)** and **(b)** only include events in the wind sector between 150 and 270° and with the stationary criteria based on the momentum flux for $N_{tot}$ and the $CO_2$ flux for $CO_2$. **(c)** Time series of the 30 min fluxes of $N_{tot}$ using the GR method in the 150–270° wind direction. Additionally, the fluxes of ship peaks are shown separately. Panel **(d)** is the same as panel **(c)** but for $CO_2$ fluxes.

sition system calculated the averages over the data collection period. It became very clear that the frequency of once per minute was too low in order to see accurate emission peaks in the ship plumes, as the duration of the plume itself was of the order of a few minutes. This was the reason for shortening the response time of the analyzers by increasing the flow rate from the nominal flow and shortening the integration time, which then made it possible to increase the data collection frequency.

As seen from Fig. S6, major factors influencing the accuracy of calculating the emission peak area are the difference in the response time between the $SO_2$ and $CO_2$ analyzers as well as the frequency of data collection. The Picarro G2301 $CO_2$ analyzer is clearly faster than the UV-fluorescence $SO_2$ analyzers, which are designed for air quality measurements to meet EU regulation requirements. Even when improvements were made to the $SO_2$ analyzer between the 2011 and 2012 campaigns, the difference in the peak width is clearly seen.

The variation between the 2011 and 2012 campaigns with respect to the calculated FSC from the ships that routinely cruise between Helsinki and Stockholm or Helsinki and Tallinn is used when estimating the uncertainty in the FSC according to Eq. (13). In Fig. 13, the relative expanded uncertainty for the FSC, $U_{FSC}$ (%), is shown as a function of the peak height concentration of $SO_2$. Calculation of $U_{FSC}$ (%) was carried out for FSC = 0.1 %, 0.5 %, and 1 %, fulfilling the regulation requirements at the time of the measurements (FSC = 1 %) but also the regulation that came into power in 2015 (Directive, 2012/33/EU) for SECA areas (FSC = 0.1 %) and for oceans (FSC = 0.5 %). The uncertainty in the FSC increases rapidly: 100 % at a peak concentration of 3 ppb for a FSC of 0.1 %, and 5 ppb for a FSC of 0.5 % TS4. When more accurate measurements are required, the peak height concentration of $SO_2$ should be clearly higher than 5 or 10 ppb for the FSC regulation of 0.1 % or 0.5 %, respectively. Considering that the highest peak concentration observed was 50 ppb, this gives an uncertainty of 18 % for FSC. At the same time, the NO peak was 100 ppb, giving an $SO_2$ analyzer response of 2 ppb or 4 % of the measured $SO_2$ with a cross-sensitivity of 2 %. According to the present regulations, FSC = 0.1 %; hence, the

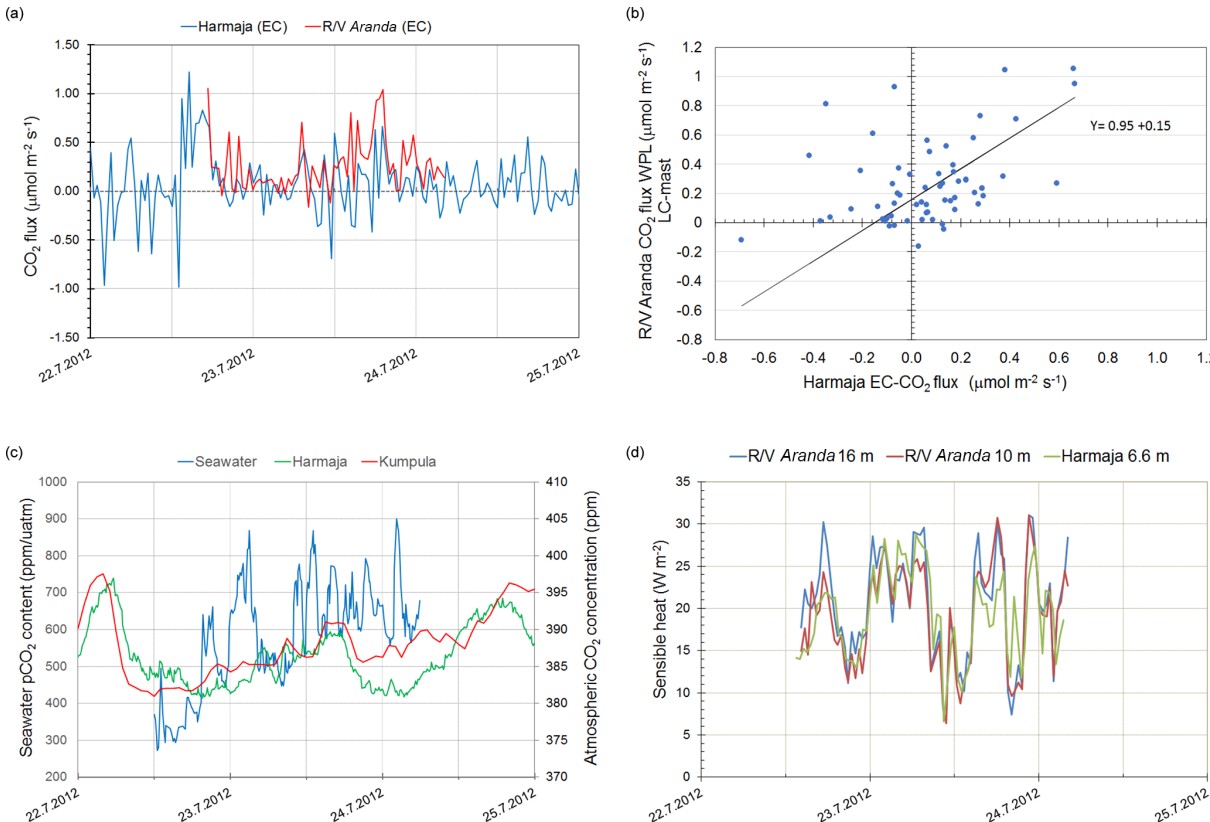

**Figure 12. (a)** Time series of $CO_2$ fluxes using the EC methods in 2012 at Harmaja and on the R/V *Aranda* in the 150–270° wind direction. **(b)** Scatterplot of the orthogonal regression analysis between EC-$CO_2$ flux measurements conducted at Harmaja and on the R/V *Aranda* in 2012. The orthogonal regression line is also shown. **(c)** Partial pressure of $CO_2$ in seawater from the R/V *Aranda*, and $CO_2$ concentrations in the atmosphere at Harmaja and at the SMEAR III urban background monitoring station in Kumpula, Helsinki. **(d)** Time series of sensible heat fluxes measured using the EC method on the R/V *Aranda* at altitudes of 16 and 10 m and in Harmaja at an altitude of 6.6 m.

highest peak concentration would be 5 ppb of $SO_2$ with an uncertainty of 25 %. The NO would still be 100 ppb, giving an $SO_2$ response of 2 ppb or 40 % of the measured $SO_2$ concentration. Correction of the cross-sensitivity for nitrogen compounds on the response of the $SO_2$ analyzer using the UV-fluorescence technique is vital.

No violations of the regulations were observed for the calculated FSC during the campaigns in 2011 and 2012 (Fig. 14). A typical FSC value of 0.4 % was obtained with an uncertainty of 15 %, i.e., $0.40 \pm 0.06$ %. Moreover, the FSC values obtained were in good agreement with the information given by the ship owners as well with our earlier results (Pirjola et al., 2014), where emissions from the same ships were studied in winter and summer campaigns in 2010 and 2011. The mobile laboratory "Sniffer" was standing at the harbor areas in Helsinki and Turku. Besides the FSCs, we also estimated the emissions factors for NO, $NO_x$, $SO_2$, $N_{tot}$, and $PM_{2.5}$, For example, the emission factors for $NO_x$ were in the range of 56 to 100 g per kilogram of fuel.

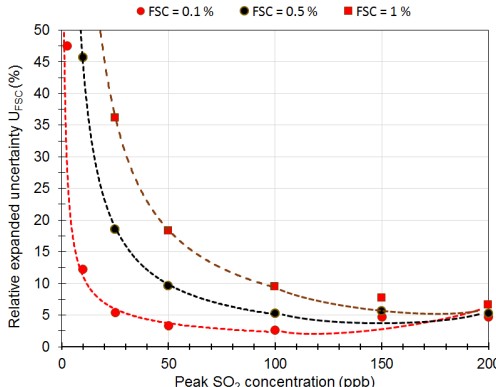

**Figure 13.** The expanded uncertainty of FSC, $U_{FSC}$ (%), as a function of the $SO_2$ peak height concentration for three FSC limits: 0.1 % (red circle), 0.5 % (black circle), and 1 % (red square).

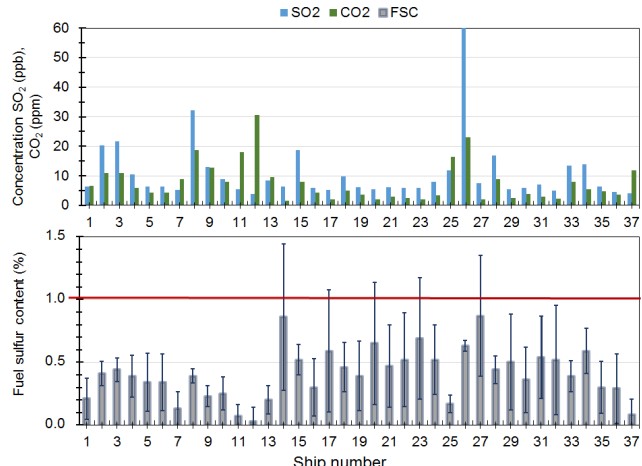

**Figure 14.** Peak height concentrations of $SO_2$ and $CO_2$ from a ship emission plume, and calculated FSCs with the expanded uncertainty $U_{FSC}$ during a short time period for the 2012 measurements. The red line represents the FSC = 1 % limit. All of the ships complied with the regulations.

## 5   Conclusions

Direct exchange of gaseous compounds and nanoparticles between the air and sea interface was studied using micrometeorological methods. The gas compounds $SO_2$, NO, $NO_2$, $O_3$, and $CO_2$ as well as the number concentration of nanoparticles, $N_{tot}$, were measured in the Baltic Sea beside the major ship routes to and from the city of Helsinki. Flux measurements in the marine environment are challenging due to meteorological conditions and topographical aspects. Filtering of data outside the footprint area and for certain wind sectors, the occurrence of swell, nonstationarity, and concentration difference between the measurement heights less than the uncertainty limit reduce the number of available data considerably. In this study, 43 % of the measured fluxes of $N_{tot}$ using the GR method and 28 % of the measured fluxes of $CO_2$ using the EC method were acceptable.

It became quite clear that no direct gas exchange across the air–sea interface (negative or positive fluxes) could be measured using the GR method. This was mostly due to the fact that the capability of the analyzers used to measure the gas concentration differences under clean coastal conditions was not sufficient. Even though the $CO_2$ flux was too small to be detected using the GR method, it could be detected with the EC method. The case was different for nanoparticles, where the observed differences in the number concentration were well above the uncertainty limit for the both ELPIs.

Both the GR and EC methods were capable of measuring the emissions from the ships. Much effort was invested in studying the transport and dispersion of single ship emissions. Different scenarios depending on the wind speed and wind direction were able to identify the following: (i) pollutants reached the footprint area and the measurement mast

or (ii) pollutants bypassed the footprint area but were seen by the measurement mast. When the mixing of the pollutants occurred well before the footprint area for the measurement mast, the measured fluxes were real. When the mixing of the pollutants from the ships was not complete, the M–O theory was violated, and the measurement results described dispersion of the pollutants

The measurements for determining the FSC were in good agreement with the information given by the ship owners. The measurement method used to determine the FSC content of marine fuel from the ambient air in connection with the identification data from AIS gives a clear demonstration of whether the regulations are respected.

The uncertainty analysis of the fluxes and the FSC was conducted according to the well-known law of propagation of errors, and following the recipes from the literature (JCGM, 2008). The uncertainty budget, which defines the sources of uncertainties and their contribution to measurement results, may be conservative; however, it can be made more accurate by selecting a better measurement technique, more capable analyzers, a homogeneous measurement site, and stationary meteorological situations. CE5

To improve the accuracy of the FSC based on measurements of the ratio of $SO_2$ to $CO_2$ from the emission peaks of ship plumes, the development of instrumentation that can simultaneously measure $SO_2$ and $CO_2$ concentrations fast enough and with the same order of response time is highly desirable.

*Data availability.* The data used in this study are available from the Zenodo data repository: https://doi.org/10.5281/zenodo/5726313 TS5 . CE6

*Supplement.* The supplement related to this article is available online at: https://doi.org/10.5194/acp-21-1-2021-supplement.

*Author contributions.* JW and LP designed the concept of the study. JW, LP, and TW performed the measurements with help from TL, JH, TT, and MM. HP and KK were responsible for the *Aranda* measurements and the aspects in the marine boundary layer. HN carried out the OpenFOAM simulations. JPJ provided the AIS data and $NO_x$ emissions from ships (Fig. 1). JW wrote the original manuscript, and LP, HP, KiKK, and JPJ edited it. All authors have read and agreed upon the published version of the paper.

*Competing interests.* The authors declare that they have no conflict of interest.

*Disclaimer.* This work reflects only the authors' views, and INEA is not responsible for any use that may be made of the information it contains.

Please note the remarks at the end of the manuscript.

*Acknowledgements.* The measurements were carried out during the SNOOP project, financed by the European Regional Development Fund, Cental Baltic INTERREG IV A Programme. The authors are very grateful to Aleksi Malinen at Metropolia University of Applied Sciences for help with the measurements and to Kaisa Lusa and Sisko Laurila at the Finnish Meteorological Institute for help with laboratory calibrations. For revision of the language, the authors are very grateful to Leena Kahma. The editor and anonymous referees are thanked for their critical and insightful but constructive comments that considerably improved the final paper. CE7

*Financial support.* This research has been supported by the European Union's Horizon 2020 program (grant no. 814893).

*Review statement.* This paper was edited by Drew Gentner and reviewed by three anonymous referees.

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

## Remarks from the language copy-editor

CE1    Please check that the meaning of your sentence is intact.
CE2    Please check that the meaning of your sentence is now correct.
CE3    Please confirm.
CE4    Please note the addition. Roman font is our house standard in this case.
CE5    Please note the slight change.
CE6    Please note the slight change.
CE7    Please note the changes.

## Remarks from the typesetter

TS1    Please note that this change will have to be approved by the editor. Please provide a short explanation that can be forwarded by us to the editor.
TS2    Not mentioned in the reference list.
TS3    Please confirm.
TS4    Please note that the changes to this paragraph will have to be approved by the editor. Please provide a short explanation that can be forwarded by us to the editor.
TS5    Please check DOI.
TS6    Please check DOI.
TS7    This reference is not mentioned in the text.