# Peer review of "Measurement report: Characterization of uncertainties of fluxes and fuel sulfur content from ship emissions at the Baltic Sea"

_Atmospheric Chemistry and Physics, 2020_

## Author Comment (AC1)

**acp-2020-1086 (Walden et al.)**
**Measurement report: Characterization of uncertainties of fluxes and**
**fuel sulfur content from ship emissions at the Baltic Sea**

**Reply to Reviewers**

**We thank the reviewers for their critical, valuable and constructive comments. We have made a major revision and replied to all the comments. Our answers below are written by blue color. The changes made in the manuscript are written below by red color, and highlighted in the manuscript by yellow color. The manuscript was significantly improved.**

**Referee 1**

Walden et al use a gradient method to investigate sea-atmosphere fluxes of various species. The detection limit of the gradient methd is not sufficient to observe exchange fluxes for most gases. The authors report particle deposition fluxes, likely originating from ship emissions. Additionally FSC is assessed. The FSC in my opinion is the most interesting part of the manuscript, why weren't concomitant NOx and particle plumes tracked, as this would seem a natural extension of the experiment.

The emissions factors for NO, $NO_x$, $SO_2$, Ntot, and $PM_{2.5}$ as well as the FSCs for the same ships were already estimated in our previous publication Pirjola et al., Atmos. Meas. Tech. 7, 249-261, 2014. Discussion of these results were added in the text.

p. 16, lines 475-479: Also the obtained FSCs were in good agreement with the information given by the ship owners, as well with our earlier results in Pirjola et al. (2014), in which the emissions from the same ships were studied in winter and summer campaigns in 2010 and 2011. The mobile laboratory Sniffer was standing at the harbour areas in Helsinki and Turku. Besides the FSCs we also estimated the emissions factors for NO, $NO_x$, $SO_2$, Ntot and $PM_{2.5}$, For example, the emission factors for NOx were in the range of 56 to 100 g (kg fuel)$^{-1}$.

As for the flux analysis a number of major issues arise when reading the paper. While presenting CFD calculations to support the measurement site setup, it is not clear whether stationarity criteria were fulfilled due to passing ships and associated plumes being advected over the site. I disagree that stationarity is primarily characterized by concentration trends, which are part of the longer wave spectrum, often filtered out by turbulence averaging intervals. There are better ways to investigate stationarity (see standard textbooks on micrometeorological data pre-processing). As such the interpretation of fluxes needs to be evaluated carefully, because many fundamental criteria often implied for flux measurements might not be fulfilled.

We considered the stationarity as between the flux estimate when the trend in the wind speed was removed and when this trend was not removed. For comparison as a criteria for stationary we used the method proposed by Foken and Wichura (1996) for fluxes of $CO_2$, heat, water vapor

and for momentum. We evaluated the observed fluxes following the QA/QC procedures presented later in the text. We replaced the stationary criteria according to the method of Foken and Wichura into the manuscript. Fundamental criteria for flux measurements stationarity of the flux measurements, the footprint area and the occurrence of swell were considered. The following text was added:

p.3, lines 68-70. The use of micrometeorologocal methods reguires criterias to fulfil for the atmospheric conditions being similar for both methods. As such stationarity of the flux measurements, the footprint area and the occurrence of swell were considered.

p. 5, lines 121 – 124. General conditions for the M-O similarity theory are horizontally homogeneous surface structure, stationary (or near stationary) condition (e.g. Foken and Wichura, 1996), constant flux layer and that the atmospheric turbulence is affecting on the vertical profiles of wind speed, potential temperature and humidity.

This might also relate to different footprints of individual levels of the gradient tower (e.g. is the lowest level even seeing the water surface or partially also influenced by the island?)

We calculated the footprint area at each of the measured height at stable, neutral and at unstable conditions according to Högström et al (2008). Based on the calculation in neutral conditions the footprint area from the lowest height accounts 0.3 % from the observed flux at the distance of 20 m from the mast (i.e. at sea line, see in Fig. 2b) while at altitude of 10.7 m the footprint area starts at 40 m from the mast. In Fig. 8b the distance where the maximum flux (red color area) is gained is presented for different heights.

We added the missing text for the analysis of the footprint area to the final manuscript.

p. 13, lines 379 - 385: The footprint area i.e. the area along the upwind where the exchange of gases and particles between the air-sea surface are expected to be a source of the measurement results, was calculated according to Högström et al. (2008). The footprint area was calculated at each of the measured height at stable, neutral and unstable conditions. Fig. 8a illustrates the relative intensity of the footprint areas in neutral conditions as a function of upwind distance from the measurement mast at instrument heights of 4.7 m, 7.2 m and 10.7 m. The cumulative relative contribution (Fig. 8b) indicates that at the lowest height (4.7 m) less than 0.3 % of the observed flux takes place at the distance of 20 m from the mast reaching 90 % at a distance of 3 km. At the height of 10.7 m, the footprint area starts at 40 m from the mast reaching 85 % at distance of 3 km.

The fact that ship plumes on the order of a few seconds were observed suggests that homogeneity and stationarity was largely not fulfilled for quantifying fluxes from ships.

The open sea sector that is studied here more intensive is to direction where the ships are approaching to or moving away from Helsinki. This means that we measure the ship plumes over the footprint area. The duration of the ship plumes varied from 3 to 7 min. The time is of the same order as the short time averages used in the stationary test by Foken and Wichura (1996). We analyzed single emission peaks from the ships. In Fig 10 is a schematic presentation on the

dispersion of ship plume (a) and the momentary plumes at different time intervals (b). In Fig. 11 d and e we present the analyzed ship peaks and compared them with the 30 min average values for $CO_2$ and $N_{tot}$.

A comparison between CO2 eddy covariance fluxes and gradient measurements is shown, but it is not indicated what QAQC criteria (e.g. u*, ICT, stationarity) were used to filter data, and how much of the original data was used for the analysis after QAQC filtering. Were storage fluxes considered?

The storage fluxes were not considered at this campaign. The site was at the sea where most of the time the turbulent mixing was the driving force for gas and particle dispersion. The following QA/QC procedures and criteria for calculated fluxes were added in the text:

p. 12, lines 360-377

Quality control (QC) and quality assurance (QA) procedures are actions that should take into account in order to improve the data quality and to make the data comparable with the similar data from other studies. Although QA and QC procedures have slightly different meanings, in this study, the quality assurance and quality control (QA/QC) procedures are considered together. The following QA/QC procedures and criteria for flux calculations were taken into account.

1. Calibration of the used analyzers for gases and particles (Section 3.3), for the GR and EC methods
2. Criteria for minimum concentration difference between the measurement heights (Fig. S2), for the GR method
3. Correction of the wind flow field around the measurement mast according to the CFD calculations (Section 4.1), for the GR method
4. Restriction to open sea, i.e. wind direction in the range of 150-270 degrees (Fig. 3a), for the GR and EC methods
5. Analysis of swell to demonstrate the validity of M-O-theory with Codes 1-3 (Section 4.1), for the GR method
6. Footprint area (homogeneous fetch area) was estimated at each of the measurement height and at neutral, stable and non-stable condition (Fig. 8), for the GR and EC methods
7. Stationarity criteria following the criteria of Foken and Wichura (Foken and Wichura 1996), for the for GR and EC methods
8. The intermittency was applied according to Mahrt et al. (1998), for the EC method
9. WPL correction due to water vapour and heat flux, for the GR and EC methods
10. Cross sensitivity of the compounds on the used analyzers, for the GR and EC methods
11. Preparation of the uncertainty budget for the measurement results, for the GR and EC methods

To that end it would also be good to quantitatively compare the two flux methods for $CO_2$, as it could help validating the gradient method. In this context I would expect to see a scatter plot and regression of both fluxes, - how well did the two methods really compare?

We agree with the comment. Unfortunately the fluxes of $CO_2$ were too small to be detected by the GR method. This can be seen from the concentration difference of the $CO_2$ between the measurement heights and shown in Fig. S2g. After the WPL correction applied to the measurement of $CO_2$ concentration the difference of CO2 did not exceed the calculated uncertainty limit. In the previous version of the manuscript this was not included into the results but at present it was corrected. We made the comparison of GR and EC methods with sensible heat, as shown in the figure below. The calculation of the sensible heat by GR method was conducted from the sea surface up the measurement height 11 m and to 15 m. The temperature difference between the measurement heights (11 and 15 m) were mostly too small to be detected.

[Figure]

Fig.1. Sensible heat by GR and EC method at Harmaja 2011. The scatter figure between the two methods is shown in the small figure.

**References**

Foken, T, Wichura, B. Tools for quality assessment of surface-based flux measurements. Agric. Forest Meteorol. 78, 83 – 105, 1996.

Högström, U., Sahlée, E., Drennan, W. M, Kahma, K. K., Smedman, A.-S., Johansson, C., Pettersson, H., Rutgersson A., Tuomi, L., Zhang, F., Johansson, M.: Momentum fluxes and wind gradients in marine boundary layer – a multi-platform study, Boreal Env. Res., 13, 475-502, 2008.

Mahrt, L., Sun, J., Blumen, W., Delany, A.C, Oncley S. Nocturnal Boundary-Layer Regimes. Boundary-Layer Meteorology 88, 255 – 278, 1998.

Pirjola L., Pajunoja A., Walden J., Jalkanen J.-P., Rönkkö T., Kousa A., Koskentalo T.: Mobile measurements of ship emissions in two harbour areas in Finland, Atmos. Meas. Tech., 7, 149 – 161, 2014.

---

## Author Comment (AC2)

acp-2020-1086 (Walden et al.)
**Measurement report: Characterization of uncertainties of fluxes and fuel sulfur content from ship emissions at the Baltic Sea**

**Reply to Reviewers**

**We thank the reviewers for their critical, valuable and constructive comments. We have made a major revision and replied to all the comments. Our answers below are written by blue color. The changes made in the manuscript are written below by red color, and highlighted in the manuscript by yellow color. The manuscript was significantly improved.**

**Referee 2**

**General**

The paper by Walden et al. focuses on flux measurement uncertainties and sulfur content from ship emissions. The measurements are conducted by gradient method for selected trace gas pollutants and supplemented by eddy covariance measurements of $CO_2$ from a 9 m mast located at a coastal site and on a research vessel 2 km SSW into the sea. Quantifying emission rates from moving ships is certainly not trivial and the number of challenges encountered by the authors is simply impressive. While the paper shows a large effort in conducting the measurements, at least in this version, I would have reservations to the data interpretations and whether they can fully capture ship emissions using the presented approach. The major result from the paper are measured FSCs from the ships all of which did not exceed the EC regulation limit. Overall, I found this paper interesting for the focus on ship emissions, but there are inconsistencies in the data and the paper shows a high potential for further analysis and more coherent presentation of the results.

**Major comments**

1.  The major question is how well the assumptions of the gradient and EC methods worked for this heterogenous coastal site. If the large short-term episode (e.g. in $SO_2$ or $CO_2$) occupies only a fraction of the flux integration period, the episodic/spike data would most likely make it nonstationary regardless of whether other micromet variables were stationary or not. The stationarity test should be conducted on each flux tracer including the $CO_2$ data.

    The stationary test according to Foken and Wichura (Agric and Forest Meteorology 78, 1996, 83 – 105) was calculated and presented in the text. See also clarification in the Referee 1 at this point.

2.  The gradient method also requires accurate measurements at two different heights. If the systematic offset between the instruments (SI Figure S1) was not corrected for, it would lead to large errors in calculated vertical GR fluxes. It is unclear how the data in Fig. S1 were used to correct/cross-calibrate the instruments when the correlation slope differs from 1.

    We calibrated the instruments for gaseous compounds according to practice used at the FMI calibration laboratory. In addition the remaining offsets were removed at the measurement site by introducing a common sample through the sampling line to all instruments. In case of particle instruments (ELPIs) cross calibration was obtained by the use of the HEPA-filter for adjusting the zero level and by common samples at the same height at actual particle concentrations. The number concentration of particles smaller than 1 µm (Ntot) from the ELPI2 was corrected with respect to the ELPI1. The text was clarified and reformulated in Supplement, and unnecessary Fig. S1 was removed.

p 9, lines 268-271: Based on the parallel measurements of the ELPIs on 30.8.2011 and 2.9. 2011 correction factors were inferred for ELPI2, separately for each stage (dN/dlogDp), and for the number concentration of particles smaller than 1 µm (Ntot) that was used for the flux calculations (more details in Supplement). All measured Ntot data from the ELPI2 were corrected accordingly.

Supplement: After the HEPA filtration tests and zero setting the same air sample was injected into both instruments at certain time frames and at different particle concentrations. The effective variance regression line between the two ELPI instruments (assuming that both instruments have the same statistical uncertainty) was used to correct the results of the ELPI2 with respect to the ELPI1. The correction coefficients were inferred for the 8 smallest stages of the ELPI2 (dN/dlogDp), and for the number concentration of particles smaller than 1 µm (Ntot). Only the Ntot concentrations (not the size distributions) from ELPI2 were used in this study. The standard and expanded uncertainties of the instruments with respect to each other was then inferred from the scatter of the Ntot results from the ELPIs (Figs. 1h and 2h). All measured Ntot data from the ELPI2 were corrected accordingly.

3. In the GR method, the authors rely on the assumption that the eddy diffusivity for heat transfer is the same as that for gas mass transfer (e.g. L.81). This could lead to large uncertainties which should be calculated independently for each chemical species. The comparison of GR and EC sensible heat fluxes could have been relatively easy and a good start in comparing the EC and GR methods.

Assumption, $K_{heat} = K_{gas}$ is general (Panofsky and Dutton, 1987). The sensible heat was calculated with both of the method and we present the results, see in figure and text in Referee 1.

4. It would have been great to see a more quantitative comparison for GR and EC methods for $CO_2$ (and for heat). However, from Fig. 9 it is clear that the $CO_2$ fluxes agreed rather poorly, where for example between 08/28 12 PM and 08/29 12 PM, the gradient data show all negative values while EC data are scattered in a broader range mostly positive values but often changing the flux sign. The relative difference between the methods for most of the measured period therefore largely exceeds the uncertainties stated in the abstract (25-36 % and 30-60 % for the GR and EC methods, respectively). For this reason, I am finding highly suspicious the exact same median value for GR and EC $CO_2$ fluxes reported in Table 1. I agree with the comment of the other referee that the scatter plot would have reflected more clearly how both methods worked. If the agreement does not work well for $CO_2$, the question is why and whether the gradient flux method was valid for $SO_2$ and other reported trace gases.

The Referee is correct and we found mistakes in our previous manuscript. We analyzed more carefully the data between 08/28 12 PM and 08/29 12 PM, and it turned out that the WPL correction for water vapor was fully responsible for the change of the sign from negative to positive values in case of $CO_2$ fluxes by EC method. Regarding to the results of CO2 fluxes by GR method we found a mistake that was corrected in the present manuscript. We calculated the minimum concentration difference of $CO_2$ in dry air which did not exceed the calculated uncertainty limit that can be detected with the used analyzers. This means that the fluxes of $CO_2$ could not be calculated. In previous manuscript, the minimum concentration difference was calculated only for wet condition and exceedances for the uncertainty limit occurred more frequently. Therefore, we removed Fig 9a and replaced it with a new figure including only the flux results of $CO_2$ by EC method.

Table 1 was corrected.

In case of applying the scatter plot between the results of $CO_2$ by GR and EC method, see our respond to the Referee 1 at this point.

5. The flux footprint contribution does not seem to be discussed. The data could give a completely different picture if the ship was outside the footprint (depositing fluxes to the site expected) compared to when the ship would be inside the footprint (emission fluxes expected). It is therefore challenging to attribute any enhancement to the ships without the knowledge of what the footprint was and how it was changing.

Text on the analysis of the footprint was added in the text. An example of analysis of the footprint is presented in Fig. 8a and b.

P 13, lines 379-388: The footprint area i.e. the area along the upwind where the exchange of gases and particles between the air-sea surface are expected to be a source of the measurement results, was calculated according to Högström et al. (2008). The footprint area was calculated at each of the measured height at stable, neutral and unstable conditions. Fig. 8a illustrates the relative intensity of the footprint area in neutral conditions as a function of upwind distance from the measurement mast at instrument heights of 4.7 m, 7.2 m and 10.7 m. The cumulative relative contribution (Fig. 8b) indicates that at the lowest height (4.7 m) less than 0.3 % of the observed flux takes place at the distance of 20 m from the mast reaching 90 % at a distance of 3 km. At the height of 10.7 m, the footprint area starts at 40 m from the mast reaching 85 % at distance of 3 km.

6. Given the moving point source within likely changing footprint (not uniform at the two heights) I am not convinced that the chosen approach was optimal for quantifying emission rates from ships. There are other methods such as wavelet analysis which could be more appropriate to measure intermittent or short-term emission episodes (e.g. Steiner et al., 2011; Misztal et al., 2014) which are not dependent on stationarity criteria.

We analyzed the transport and dispersion of a ship plume using the data from the emission measurement conducted at know regular cruising ships between Helsinki and Stockholm and applying the dispersion models. In Fig 10 a schematic layout on the dispersion of the ship emission (a) and an example of the transport and dispersion of the ship emission at known condition at August 28 with different time steps reaching the footprint area is presented (b).

7. I could not find it in the main text and SI, so I am curious if the data were subjected to coordinate rotation and how close to zero was the average vertical wind speed w? A small tilt of the sonic anemometer could greatly skew the flux data.

The coordinate correction for the vertical wind speed, $w$ was corrected in the Sonic signal during the calculation. In the direction to open sea, the wind angle was tilted 8 ± 6 Degree.

8. There is no mention about how the lag time was derived for each integration period or if a constant value was used. I am particularly concerned about the potentially incorrect lag time because the $CO_2$ flux was changing sign from one period to the other like, for example, from 28 to 30 Aug (Fig. 9a). It would be great to see how a peak in the covariance function looked like and if the lag time was stable.

The synchronization for the signal from Licor and the sonic wind speed was corrected during the measurements. The lag time was monitored over the measurement campaign. The change of sign during the period from 28 to 30 Aug was analyzed more carefully and as mentioned at Q4, the correction of water vapor was mainly responsible for the change of sign. This can be demonstrated according to:

According to Webb et al. (1980) the covariance term (w's') can be expressed as

$$w's' = (M_c/M_a)[w'CO_{2,w}'/(1 - H_2O) + w'H_2O' \cdot CO_2/(1-H_2O)^2]$$

where $M_c$ and $M_a$ are the mole mass of $CO_2$ and dry air, $w'CO_{2,w}'$ is the raw CO2 flux in (ppm m $s^{-1}$), $H_2O$ is the water content in (mmol $mol^{-1}$), $w'H_2O'$ is the raw water flux (mmol $mol^{-1}$ m $s^{-1}$) and $CO_2$ is the concentration (wet). We divide the correction into two terms on the right side, terms 1 and 2 as well as their sum 1+2, shown in the Fig.2a. The raw flux of CO2 ( $= w'CO_2'$) and the WPL-corrected w's' (= WPL 1 + 2 terms) on the right, Fig 2b.

[Figure]

Fig 2. The WPL correction presented in two term and their sum (a) and for comparison the raw $CO_2$ flux and the WPL correction (b).

9. The data quality control is not presented clearly. It would be great to know what criteria were used and which data were actually rejected. For instance, Figures 8 and 9 show the data for when M-O theory was not fulfilled. If it is important to show these low-quality data they could be shown in grey so it is clear that they were rejected and are not distracting from observing potentially good data.

   A text for describing the QA/QC procedures was added into the text, see the comments by the Referee 1. We changes Figs 8 and 9 accordingly.

10. Conclusions lack the main take-home messages. Practically entire conclusions are spent on emphasizing high uncertainties and challenges and not pointing out the main results or findings. Was the goal to say that the methods did not work at all or that they might potentially work with some improvements? Including the major findings based on the valid data (FSCs?) and further analysis of the remaining data (especially NOx) could significantly improve the manuscript.

   Section Conclusions was rewritten.

**Specific comments**

10. What inlet was used for sampling ultra fine particles?

   The following information was added.

   p 9, line 240: For particle sampling stainless steel tubes with an outer diameter of 12 mm were used.

11. SI Figure S2 shows how uncertainty increases closer to the detection limit which is a nice demonstration. However, it is unclear how the data below the detection limit were treated. I suggest to consult Helsel (1990).

The average concentration of 30 min for SO2 and NO were mostly close to or below the detection limit, and we should not conduct any calculations (subtraction) with this kind of data. Therefore we discarded the results shown in Figs. S2a,d (now Fig. S1 a,d). (Tarkemmin ajatellen näin pitäisi tehdä. En suosi, että muutetaan arvot alle DL:n esimerkiksi DL/2 ja lasketaan sitten erotuksia tms.) In case of other gases concentrations were above the DL, especially in case of O3 and CO2. Calculations for fluxes were conducted if the difference in concentrations between the sampling heights exceeded the uncertainty limit, i.e. repeatability (see definition e.g. by EN 14211) of instruments at the measured concentration.

12. What was the message the multipanel Figure S3 was meant to come across? Is it suggested that the absolute uncertainties exceeded almost all the data values? It could perhaps be clearer to show the relative uncertainties as shaded areas.

If the concentration difference between the two heights was less than the repeatability of the instruments at that concentration, the results were discarded. Unfortunately, this was the case for most of the time for all gases. Only the concentration differences for $N_{tot}$ exceeded clearly the uncertainty limit enabling the calculation of the Ntot fluxes by the GR method. Fig. S3, now Fig. S2, was improved as suggested by the Referee.

13. In the uncertainty budget, I would suggest the authors describe the systematic and random errors as well as the treatment of data below the detection limit. It is unclear if the data have been corrected for the systematic error.

Calibration of the response of instruments and the cross-calibration due to sampling were systematic errors and corrected. Instead, the variation for temperature, pressure and humidity in the atmosphere influencing on the respond of the instruments were treated as random errors and included into the uncertainty budget (shown Fig. S2, now Fig. S1). We included a clarification for the text in case of how the data were corrected for systematic errors, eqs. 11-13 and Table S1. We added in eq. 12 the standard uncertainty for wind speed. Additionally, we also considered the statistical error of an EC estimate.

p. 7, lines 180-183: The uncertainty sources that contribute to the uncertainty of the flux results by the GR method are systematic and random in nature. Calibration of the response of all instruments, correction of the humidity for $CO_2$ analyzers and cross-calibration due to sampling tubes of the analyzers are systematic errors. All the uncertainty sources (systematic and random) that contribute to the results need to be corrected.

p. 13, lines 393-399: The uncertainty sources of the measurement results for fluxes by the gradient method are presented in more detail in Supplementary Table S1. To estimate the uncertainty of the momentum flux and $CO_2$ flux measurements by the EC method we calculated the expected statistical variability using the Co-spectrum. For the momentum flux it was 20 %, and for the $CO_2$-flux 30 %. This wide uncertainty range is typical in real meteorological situations and explains the scatter of the EC estimates in e.g. Fig. 12. The analysis of uncertainty follows the guideline provided by the Joint Committee for Guides in Metrology (JCGM, 2008). Based on the analysis, the relative expanded uncertainties for the flux measurements of $CO_2$ and nanoparticles are presented in Table 1 at stationary meteorological conditions.

14. Eq. 10, the value of the 0.232 multiplier seems somewhat off when using the emission factor from Petzold et al. Was a different EF value used instead?

We have used $EF_{CO2}$ = 3107 g/(kg fuel) from Petzold and Equation (4) from Pirjola et al., 2014. The coefficient should be 0.226 ~ 0.23. This value was used for the FSC calculations. Equation (10) was corrected accordingly.

15. Figure 1, poor resolution, I could not read the text.

The resolution of Figure 1 was improved.

16. Figure 3, panel a) low resolution CFD figure, I could not read the legend. It would be useful to add in the text how exactly CFD was used to correct the data and if it was a constant or time-dependent correction.

We clarified the text and figure caption. Also the resolution of Fig. 3a was improved.

p. 11, lines 309-314: The airflow around the shoreline and the measuring structure was modelled using steady, incompressible, single-phase potential flow. The simulation covered a 80 m long, 40 m wide and 30 m high rectangular box around the measurement area. Fig. 3a illustrates the calculated wind field isopleths at a wind speed of 9 m/s over the open sea area, and it shows how the flow field is disturbed around the measurement mast. Based on the calculated isoplets we determined for each measurement height the corresponding height over the open sea. The actual wind speed probe heights are shown at the measurement mast on the right and their projected heights over open sea on the left.

Figure 3. a) The actual measurement heights were reduced to corresponding heights over the open sea surface utilizing calculated isopleths from flow dynamics program. The actual wind speed probe heights at the measurement mast are shown on the right and their projected heights over open sea on the left.

17. Figure 5, these trajectories show long-range transport. Can these be zoomed to the measurement site?

Unfortunately, we were not able to zoom the trajectories to the measurement site. However, as seen from Fig. 1 the Harmaja island is close to the center of Helsinki, around 6 km.

18. Figure S1. Make x and y axes consistent. Show the 1:1 line. Were these data used to correct the instruments? How?

This figure was removed since the size distribution data from ELPI2 was not used in this study. For the flux calculations we only needed the corrected Ntot concentrations.

19. I am surprised that the uncertainty is shown in Table S1 for the nonstationary periods as I do not think it is meaningful. The nonstationary periods should have been rejected. Did the CFD calculation correct only for the horizontal wind speed?

The Referee is right. The nonstationary periods were rejected. In the CFD calculations the vertical wind speed is constant.

20. What was the frequency distribution of $CO_2$ fluxes (FFT spectrum)? As the flux data collection was conducted only at 1 Hz (L. 275), were the data corrected for high frequency losses?

The data were corrected at 10 Hz frequency. In the early version, there was a mistake which was now corrected.

**Other corrections and changes made to manuscript:**

p. 1 line 12-14: Fluxes of gaseous compounds and nanoparticles were studied by micrometeorological methods at Harmaja in the Baltic Sea. The measurement site situated by the ship route to and from the city of Helsinki

p. 1, line20 – 21. No clear fluxes across the air-sea nor sea-air interface were observed for $SO_2$, NO, $NO_2$, $NO_x$ (= NO + $NO_2$), $O_3$, and $CO_2$ by the GR method.

p. 2, lines38 – 39: In the Batic Sea few measurement facilities to measure the gas exchange between the sea-air interface by micrometeorological methods has been set up (Smedman et al. 1999, Honkanen et al. 2018).

p. 2, lines47 – 49: The goal of this study was (i) to measure the gas and nanoparticle exchange between the sea-air interface in marine coastal environment close to the ship routes, (ii) to study the transport and dispersion of the ship plume to the footprint area, (iii) to define the FSC from the ship emission plumes and (iv) to characterise the uncertainty sources of the measurement results.

p. 3, lines 83 – 84: where $F_c$ is the flux of the scalar quantity $c$, $K_c$ is the eddy diffusivity of $c$; $c$ means here the gas compounds and particles. The gradient $\partial c/\partial z$ describes the mean concentration of c in the vertical direction $z$

p. 5, lines 133 – 135: where $\rho_a$ is the density of dry air, and $c'$ is the measured molar fraction of $CO_2$ ($\mu$mol $mol^{-1}$). The commonly used infra-red analyzers measure the concentration of $CO_2$ in air normally in wet condition unless an air drier is used in the sampling tube. The widely used method to correct the fluctuations of water vapour and heat is the so called WPL method proposed by Webb et al. (1980), which is applied in this study.

p.13, lines 389 – 390: Cross sensitivity of the compounds (e.g. water vapour) on the response of the used analyzers are included into the uncertainty budget or corrected directly on the results, see in Fig. S1.

p.13, lines 393 – 399: The uncertainty sources of the measurement results for fluxes by the gradient method are presented in more detail in Supplementary Table S1. To estimate the uncertainty of the momentum flux and $CO_2$ flux measurements by the EC method we calculated the expected statistical variability using the Co-spectrum. For the momentum flux it was 20 %, and for the $CO_2$-flux 30 %. This wide uncertainty range is typical in real meteorological situations and explains the scatter of the EC estimates in e.g. Fig. 12. The analysis of uncertainty follows the guideline provided by the Joint Committee for Guides in Metrology (JCGM, 2008). Based on the analysis, the relative expanded uncertainties for the flux measurements of $CO_2$ and nanoparticles are presented in Table 1 at stationary meteorological conditions.

p. 14, lines 404 – 407: Wind speed and friction velocity in Fig. 9c show a clear dependence on the wind direction. A linear relationship between the average wind speed and the friction velocity is seen in the sectors where the wind arrives over an open sea area, whereas non-linear behaviour is seen towards the northern sector (345° to 45°), where there are more obstacles.

p. 14, lines: 409 – 419: Dispersion of a ship plume is schematically presented in Fig. 10a. The black curves cover the area of the pollutants' dispersion, where the upper line is limited by the boundary layer while the lower curve hits the sea surface forming a new boundary layer. The fluxes from the ship can be measured if the measurement instrument is inside the new boundary layer where the footprint area exists. As an example, Fig. 10b illustrates the momentary plumes at the sea surface for a ship operating to the city of Helsinki and passing Harmaja Island with a speed of 21.5 kn (~11 m $s^{-1}$). The arrows show how the apparent plume is generated in the (u,v)-coordinate system, and where the pollutants transporting to the footprint area come from. The wind speed was 11 m $s^{-1}$ and wind direction 210° as in the afternoon and evening of 28 August (Fig. 4c). The momentary plume figures are shown after 15, 23, 30 and 37 min the start, and the plume concentration gradients decrease as the plume moves further. At the footprint area the gradient is really small indicating horizontally homogeneous situation. If additionally, the stationary criteria for heat, water vapour and momentum are valid, the momentary vertical gradients give the momentary flux.

p. 15, lines 428 – 434: The fluxes of $CO_2$ (EC) and $N_{tot}$ (GR) are presented as a function of wind direction in Fig. 11a. The fluxes were averaged over the wind sectors of 10 degrees, but no other restrictions included. Fig. 11b illustrates the time series for the $CO_2$ flux by the EC method, and Fig. 11c for the $N_{tot}$ flux by the GR method along with the uncertainties. Only the fluxes that fulfilled the stationary criteria with no swell in the wind sector of 150-270 degrees were taken into account. It can be observed from Fig. 11a that the $CO_2$ fluxes show only a weak dependence on the wind direction except in the northern sector due to the city of Helsinki (see also Fig. 7), whereas the negative $N_{tot}$ fluxes appear on the wind sectors containing ship routes (150-270°).

p. 15, lines 435 – 445: The WPL-correction to the $CO_2$ flux by the EC method corrects wet air into the dry air and for water flux. Due to the damping of temperature fluctuations in the long sample tube the WPL correction for heat flux was insignificant (Rannik et al. 1997). During the period when the air masses were moving from the Atlantic (during 28.8 to 31.8), the correction due to water flux was dominating and caused a positive offset to the $CO_2$ flux. The observed $CO_2$ fluxes are also in line with the previous measurements (Honkanen et al. 2018) at the Utö island in the Baltic Sea.

p. 15, 454 – 455: 12a. The sensible heat fluxes (Fig. 12c) were calculated by the EC method in 2012 campaign at Harmaja and at two heights (10 m and 16 m above the sea level) at R/V Aranda showing good agreement with each other.

p. 15, lines: Discrepancies between the measurement results by the GR and EC methods have been reported in the literature (Mycklebust et al. 2008, Muller et al. 2009) for $CO_2$ and for $O_3$, but no systematic reason has been found.

p. 17, lines 498 – 513: Flux measurements in marine environment are challenging due to meteorological conditions and topographical aspects. Filtering of data outside the footprint area and for certain wind sector, occurrence of swell, non-stationarity, and the concentration difference between the measurement heights lower than the uncertainty limit reduces the number of available data considerably. In this study, fluxes of 43 % for $N_{tot}$ by the GR method and 28 % for $CO_2$ by the EC method of all measurement results were acceptable.

It became quite clear that no direct gas exchange across the air-sea interface, negative or positive fluxes, could be measured by the GR method. Mostly because the capability of the used analyzers to measure the gas concentration differences under clean coastal conditions was not sufficient. Even though the $CO_2$ flux was too small to be detected with the GR method, it could be detected by the EC method. The case was different for nanoparticles where the observed differences of the number concentration were well above the uncertainty limit for the both ELPIs.

Both of the GR and EC methods were capable for measuring the emissions from the ships. Much effort was laid down on studying the transport and dispersion of single ship emissions. Different scenarios depending on the wind speed and wind direction were able to identify: (i) pollutants have reached the footprint area and the measurement mast (ii) the pollutants are bypassed the footprint area but catched by the measurement mast. When the mixing of the pollutants occurred well before the footprint area for the measurement mast the measured fluxes were real. When the mixing of the pollutants from the ships was not complete, violation of the M-O theory occurred and the measurement results described dispersion of the pollutants

**Revised figures:**

p. 23. New Fig 1.

p. 31. New Fig 8. a) Flux footprint areas at neutral stability seen by the $CO_2$ instruments at 10.7 and 7.2 altitudes, and by the ELPIs at 7.2 and 4.7 m altitudes. X-axis refers to the upwind distance from the instruments, and the colour bar to the relative intensity of the sea surface area to the flux. Figure b) shows the cumulative relative contribution

p. 32 Fig 8 number changed into Fig 9.

p. 34, lines 762 – 767: Figure 11. a) Fluxes of $CO_2$ by the EC method and $N_{tot}$ by the GR method are presented as a function of wind direction. No criteria has been included. Time series of 30 min fluxes for $CO_2$ ($\square$mol m$^{-2}$ s$^{-1}$) by the EC methods (b) in all wave conditions (codes 1 to 3), and $N_{tot}$ (cm$^{-3}$ m s$^{-1}$) by the GR method (c) with no swell (codes 1-2), along with the uncertainties. The data in (b) and (c) include only events in the wind sector between 150° and 270° and with the stationary criteria based on the momentum flux for Ntot and the $CO_2$ flux for $CO_2$. d) Time series of the 30 min fluxes of Ntot by the GR method in the wind direction of 150-270 °. Additionally, the fluxes of ship peaks are shown separately. e) The same as d) but for $CO_2$ fluxes.

p.35 lines 771 – 774: New number of Fig 12. a) Time series of $CO_2$ fluxes by the EC methods in 2012 at Harmaja and at the R/V Aranda in the wind direction of 150-270°. b) Partial pressure of $CO_2$ in the seawater from R/V Aranda, and $CO_2$ concentrations in the atmosphere at Harmaja and at the urban background monitoring station SMEAR III in Kumpula, Helsinki. c) Time series of sensible heat fluxes measured by the EC method at Aranda at 16 and 10 m altitudes and in Harmaja at 6.6 m altitude.

p. 36, line 786: New number Fig 13.

p. 37, line 791: New number Fig 14.

**References added in**

p. 19, lines 568-569: Foken, T, Wichura, B. Tools for quality assessment of surface-based flux measurements. Agric. Forest Meteorol. 78, 83 – 105, 1996

p. 19, lines 580 – 581: Honkanen M., Tuovinen J-P., Laurila T., Mäkelä T., Hatakka J., Kielosto S., Laakso L. Measuring turbulent CO2 fluxes with closed-path gas analyzer in marine environment. Atmos. Meas. Tech. 11, 5335 – 5350, 2018.

p. 20, lines 616 – 617: Mahrt, L., Sun, J., Blumen, W., Delany, A.C, Oncley S. Nocturnal Boundary-Layer Regimes. Boundary-Layer Meteorology 88, 255 – 278, 1998.

p. 20, lines 625 – 628: Muller J. B. A., Coyle M., Fowler D., Gallagher M., Nemitz E. G., Persival C. J. Comparison of ozone fluxes over grassland by gradient and eddy covariance technique. Atmos. Sci. let. 10, 164 – 169, 2009.

p. 20, lines 627-629: Myklebust M. C., Hipps L. E., Ryel R. J. Comparison of eddy covariance, chamber, and gradient methods of measuring soil CO2 efflux in an annual semi-arid grass, Bromus tectorum. Agric. For. Meteorol. 148, 1894 – 1907, 2008.

p. 21, lines 641 – 642: Rannik, U., Vesala, T., and Keskinen, R. On the dampingof temperature fluctuations in a circular tube relevant to the eddy covariance measurement technique. J. Geophys. Res. 102, 12789 – 12794, 1997.

**Other References**

Helsel, D.R.: Less than obvious-statistical treatment of data below the detection limit. Environmental science & technology, 24(12), pp.1766-1774, 1990.

Misztal, P. K., Karl, T., Weber, R., Jonsson, H. H., Guenther, A. B., and Goldstein, A. H.: Airborne flux measurements of biogenic isoprene over California, Atmos. Chem. Phys., 14, 10631–10647, https://doi.org/10.5194/acp-14-10631-2014, 2014.

Steiner, A. L., Pressley, S. N., Botros, A., Jones, E., Chung, S. H., and Edburg, S. L.: Analysis of coherent structures and atmosphere-canopy coupling strength during the CABINEX field campaign, Atmos. Chem. Phys., 11, 11921–11936, https://doi.org/10.5194/acp-11-11921-2011, 2011.

Foken, T, Wichura, B. Tools for quality assessment of surface-based flux measurements. Agric. Forest Meteorol. 78, 83 – 105, 1996.

Panofsky, H. A., Dutton, J. A.: Atmospheric turbulence - Models and methods for engineering applications. John Wiley & Sons, Inc., New York, USA, 397 p, 1987.

Webb, E.K., Pearman, G. I., Leuning R.: Correction of flux measurements for density effects due to heat and water vapour transfer, Quart. J. Met. Soc., 106, 85 – 100, 1980.

---

## Author Response (AR2)

**acp-2020-1086 (Walden et al.)**
**Measurement report: Characterization of uncertainties of fluxes and**
**fuel sulfur content from ship emissions at the Baltic Sea**

**Reply to Editor**

**We thank the Editor for the positive feedback. We have made a minor revision and replied to all the comments. Our answers below are written by** blue color. **Additionally, language was checked and some clarifications were added. The changes made in the manuscript are written below by** red color, **and highlighted in the manuscript by yellow color.**

Technical Comments:
page 5, line 135: Marine EC CO2 measurements are difficult. The methods are important, and measured fluxes can be off by orders of magnitude and of the wrong sign if methods are not optimal. Most EC measurements at sea that do not actively remove water vapor fluctuations from the air stream have been discredited. This manuscript does not provide evidence that their EC CO2 flux measurements are sound. How large was the Webb correction term compared to the actual CO2 flux (they say it is small but did not say how small)?

We agree that marine EC CO2 measurements are complicated, see also the answer to Editor's comments (below) concerning page 10, line 295.
We added a new figure in the SI (Fig. S5) to show the WPL correction for water vapor with respect to uncorrected $CO_2$ flux in both campaigns at Harmaja 2011 and Harmaja 2012. Correction for temperature was omitted as described in the text (p. 15, line 440). As can be seen, during the period of 28.8 to 31.8 the WPL correction for water vapor is significant changing even the sign of the $CO_2$ flux. At Harmaja 2012 the influence of the WPL correction was clearly less than in the campaign in 2011. In the attached figure the period when the air masses were arriving from the Atlantic Ocean, i.e. 28.8. to 31.8., the latent heat flux increased 3 to 6 times compared to rest of the period and explained the change of the sign of the $CO_2$ flux.

The following test was added in:
page 15, lines 458-461. The WPL correction performed to both of the data in 2011 and 2012 was small in general, see Fig S5. In the 2011 the sign of the flux was changed after correction of WPL during the period between 28.8 and 31.8.

page 8, line 235: how did the EC CO2 fluxes from the LI7000 compare with the Picarro?

The EC $CO_2$ fluxes were measured only by LI7000. We used Picarro G2301 for gradient technique together with LI7000 since the model G2301 is not fast enough for EC technique. This was mentioned in Section 3.2, page. 8, lines 238-240.

page 10, line 295: R/V Aranda LI7200 EC CO2 measurements were made on an undried air stream, relying instead on the Webb correction. I'm not aware of credible EC CO2 flux measurements made using an undried LI7200. There are several papers on this topic now (eg, Landwehr et al 2014, and others).

In our paper Sahlée et al. (2011) we compared the WPL corrected open path to the closed path sensor mounted on the same mast on R/V Aranda. The results were good for the small fluxes, under $0.5 \times 10^{-3}$ mmol/m3*m/s. In the present study the LI-7200 fluxes were under that limit. According to Landwehr et al 2014 the flux estimates between the open path and closed path instruments agree well when the latent heat is less than 7 W m$^{-2}$. During the campaign R/V Aranda was near Harmaja in 2012 the latent heat flux was lower than this limit most of the time, see attached figure.

[Figure]

Figure. Latent heat flux during Harmaja 2012 campaign. The small figure presents the latent heat flux during the whole measurement campaign and the bigger figure during the joint measurements by R/V Aranda.

As pointed out by the reviewer (comment on Fig 12b), the $pCO_2$ difference was large during the R/V Aranda measurements, well over the flux detection limits suggested by Blomquist et al. (2014). Also the ship movements inside the archipelago were small and could not be detected from the LI-7200 timeseries (Miller et al. 2010).

However, there is a source of uncertainty in the R/V Aranda $CO_2$ flux measurements: the flow rate was low (10 slpm), and the fluxes might be underestimated. The fluxes in the beginning of the period agreed well with the open path LI-7500 at the same height on the mast which gives confidence on the LI-72000 measurements. After rain showers in the middle of the period, the LI-7500 showed much higher values that might be due the wet lens on the LI-7500. The LI-7200 is better shielded from the rain.

The magnitude of the R/V Aranda LI-7200 fluxes also compared rather well to the $CO_2$ fluxes measured at Harmaja where the measuring system was comparable to LI-7200, using humid air.

We added a sentence into:
page.15, lines 453-457: The corrections to $CO_2$ fluxes measured from humid air samples have been criticized to be insufficient (e.g. Landwehr et al. 2014). The fluxes in this study mainly fulfilled the criteria presented in Miller et al. (2010), Sahlée et al. (2011), Blomquist et al. (2014) and Landwehr et al. (2014). The $CO_2$ fluxes measured on R/V Aranda might be underestimated because the flow rate was too small, 10 slpm. No systematic differences were observed when the LI-7200 CO2 fluxes were compared to the LI-7500 on the ship mast and also to the measurements made at Harmaja.

page 13, line 395: "we calculated the expected statistical variability using the Co-spectrum". What does this mean - haven't heard of this technique before?

The statistical variability of the covariance (which equals the integral of the Co-spectrum) depends on the shape of the Co-spectrum. A white Co-spectrum implicates smaller statistical variability for the covariance than a peaked Co-spectrum. In our estimates we have taken the observed shape of the $CO_2$ Co-spectrum into account when calculating the estimate of the statistical variability of the covariance.

Comments on Figures

General comment on figures: many of them were difficult to read. Fonts were small, data points were small, color choices were difficult to discern. I had to have figures zoomed on a 27" monitor to interpret.

Fonts and curve thicknesses were enlarged in most of the figures.

Fig 1: suggest turning off satelite layer because large features that may be the bathymetry are easily construed as land. There is no scale shown on this map.

We improved the figure.

Fig 2. The yellow arrow hardly shows up - suggest making it more prominent.

The color of the arrow is now red and more prominent.

Fig. 3a. I do not have any height scales shown on the left for right axes. There appears to be a missing/broken streamline at the height of the second lowest probe height on the simulated mast on the right side. Why does the right side mast extend below the soil line to the same depth as the left side mast? Shouldn't it terminate just below the lowest streamline? What is the grey stippled shading near the bottom of the figure represent - land? What is the upside-down U-shaped object

to the right of the right side mast?

We made a new figure to improve the quality of the figure and to make it clearer.

Fig 3b. The wind profiles are extrapolated from 6 m to the surface. How is this being done - they do not all appear to asymptote to (0,0)? Are you assuming some sort of roughness? My preference would be to stick to the data and not extrapolate, so profiles would extend 6-16 m.

Corrected.

Fig 3c What sort of fit is used to generate the dashed profiles? I would eliminate those curves and instead just connect the measured data points, similar to Fig 3a.

Corrected.

Fig 4. First 3 days 8/25-8/28 appear most 'active' in terms of the pollutant spikes or features in these time series in panels a,b,e. Then a precipitous drop at 8/28 around 1200 (precip?), after which fewer pollutant spikes.

No precipitation occurred. The reason for lower concentrations can be seen from Fig. 4c and Fig. 5; after noon 8/28 the western wind from Atlantic via Baltic Sea dominated carrying very clean air to Helsinki. For that reason it is important to show the trajectories in Fig. 5.

You could combine panels c and d using a little creativity - not clear why a separate panel is needed.

Panels c and d were merged.

Fig 5. How useful are these compared to the local wind patterns since the focus is on local emission sources which probably dominate the pollutant signals measured.

Please, see our response concerning Fig. 4.

Fig 8. panel a) no x-axis label provided. panel b) no x-axis label or units provided. The location of the shoreline could be added to these panels - it would be close to the right axis. The footprints are shown for neutral stability in both panels. Figure 9a shows that many (most?) periods were not neutrally stable (eg, abs(z/L)>0.1). Maybe one of these panels could be used to show the impact of non-neutral conditions on the footprint - both stable and unstable.

The labels, units and the shoreline were added to Fig. 8. The footprints in case of stable and unstable stratification are now shown in the supplement, Fig. S3.

Fig 11. panel a) what is the utility of the EC $CO_2$ flux measurements? Are these values

reasonable? For example, what was pco2 in the seawater in the footprint? This could be used to assess whether the EC CO2 flux measurements are reasonable. Alternatively, they could assume a gas exchange coefficient (eg use Wanninkhof 2014) and compute what pCO2 was in the footprint. Are these pCO2 values reasonable?

The $CO_2$ fluxes were studied as a part of the ship emission study. However, Fig. 11a where the $CO_2$ and Ntot fluxes were presented as a function of wind direction was removed.
As stated in the text the results of the fluxes obtained at Harmaja during 2011 and 2012 were similar in magnitude than by Honkanen et al. (2018). Unfortunately, we did not have measurements of the content of $CO_2$ in the seawater in 2011. This was one of the reason to repeat the measurements in 2012 together with R/V Aranda where those measurements exist. It would be interesting to study how well the bulk models for the $CO_2$ transfer based on gradient data from open ocean describe the conditions in the coastal areas and archipelago. The transfer is controlled by turbulence from different sources: the formulas relying on wind speed as a proxy for the turbulence may not give the correct answers since the conditions in coastal areas differ from those in the open ocean. This would/will be an interesting study of its own, though.

Panel b shows that most EC CO2 fluxes are positive, so that surface water concentrations should be higher than air side. But there are several points (maybe 10%) where EC CO2 flux is negative. Did surface pCO2 concentrations decrease at these times, or were fluxes countergradient? (Ok - it looks like they are suggesting the noisier points are associated with ship-induced spikes in CO2 flux).

Direct measurements of the content of $CO_2$ in seawater were not available at the time of the measurement campaign.

Fig 12 panel a) comparison between EC CO2 flux from Harmaja and Aranda could be shown better. Add lines to the timeseries similar to Fig 12c. And add a scatter plot with Aranda on one axis and Harmaja on the other axis

Fig. 12a was corrected as suggested by the Editor.
The scatter plot was added, now Fig. 12b. The following text was added to:
page 15, lines 465-467: The scatter plot of the $CO_2$ fluxes at Harmaja and on R/V Aranda is presented in Fig. 12b. The large scatter is most probably due to the different locations: the measurements of pCO$_2$ (Fig. 12c) suggest that the pCO$_2$ was not spatially homogeneous. The estimated statistical variability is large and also contributes to the scatter.

panel b) Water pCO2 values of 900 ppm are large! That should provide a good flux signal. An interesting comparison would be to compute pCO2 using the EC CO2 flux measurements from Harmaja and a gas exchange coefficient (as suggested in comment on Fig 11), and to plot that with measured Aranda pCO2 concentration.

See the answer for the comments concerning Fig. 11a.

Follow-up points on your responses to referee #1 comments:
- Given their question about storage fluxes (and your response), should the fact that storage fluxes were not considered be added to the manuscript somewhere?

The following text was added to page 13, lines 387-388:
The storage fluxes were not considered at this campaign. The site was at the sea where most of the time the turbulent mixing was the driving force for gas and particle dispersion.

- Is it valuable to include Fig 1 from your responses to referee #1 in the supplement in case other readers have the same question?

The figure was added to the supplement, now Fig. S3, and the following text was added to page 14, lines 434--437.
We also made the comparison of GR and EC methods with sensible heat (Fig. S3). Clear correlation between the methods can be observed if the calculation of the sensible heat by GR method was conducted from the sea surface up the measurement height 11 m and to 15 m. However, the temperature difference between the measurement heights (11 and 15 m) were mostly too small to be detected and no flux could be calculated.

References:

Blomquist B. W., Huebert, B. J., Fairall, C. W., Bariteau, L., Edson, J. B., Hare, J. E., McGillis, W. R.: Advances in Air-Sea $CO_2$ Flux Measurement by Eddy Correlation, Boundary-Layer Meteorol. 152,245-276. DOI 10.1007/s10546-014-9926-2, 2014

Landwehr, S., Miller, S. D., Smith, M. J., Saltzman, E. S., Ward, B.: Analysis of the PKT correction for direct $CO_2$ flux measurements over the ocean, Atmos. Chem. Phys. 14, 3361-3372. doi:10.5194/acp-14-3361-2014, 2014

Miller, S. D., Maradino, C., Saltzman, E. S.: Ship-based measurement of air-sea CO2 exchange by eddy covariance – J. Geophys. Res. 115, D02304, doi:10.1029/2009JD012193, 2010.

Sahlée, E., Kahma, K., Pettersson, H., Drennan, W. M.: Damping of humidity fluctuations in a closed-path system.- In In Gas Transfer at Water Surfaces 6 (Eds. Komori S., McGillis W. and Kurose R.), Kyoto University Press, Kyoto, Japan, 516-523, 2011.

---

## Author Response (AR3)

**acp-2020-1086 (Walden et al.)**
**Measurement report: Characterization of uncertainties of fluxes and fuel sulfur content from ship emissions at the Baltic Sea**

**Reply to Editor**

**We thank the Editor for the positive feedback and bringing forth the points that really needed clarification. Our answers below are written by blue color. The changes made in the manuscript are written below by red color.**

Thank you for the thoughtful responses and edits. After reviewing the responses, there are only 2 small questions before acceptance.

Editor question: Do you feel the following point is adequately conveyed in the manuscript to avoid similar confusion from future readers of the paper (and referenced to help any readers find the details)?

"page 13, line 395: "we calculated the expected statistical variability using the Co-spectrum". What does this mean - haven't heard of this technique before?

The following clarification was added on page 13-14, lines 399-403:
The statistical variability of the covariance (which equals the integral of the Co-spectrum) depends on the shape of the Co-spectrum. A white Co-spectrum implies smaller statistical variability for the covariance than a peaked Co-spectrum. In our estimates we have taken the observed shape of the $CO_2$ Co-spectrum into account when calculating the estimate of the statistical variability of the covariance, see e.g. Bendat and Piersol, Ch. 7 (2010).

Bendat, J.S. and Piersol, A.G.: Random Data; Analysis and Measurement Procedures. (4th ed.). John Wiley and Sons, New York, 2010.

Editor question: Is the following reason related to the shift in wind directions reason conveyed in the text for future readers?

"Fig 4. First 3 days 8/25-8/28 appear most 'active' in terms of the pollutant spikes or features in these time series in panels a,b,e. Then a precipitous drop at 8/28 around 1200 (precip?), after which fewer pollutant spikes.

We clarified the text on page 12, lines 341-353:

The 96 h backward trajectory analysis of Flextra by NILU (Stohl et al., 1995) showed that in the measurement period before the noon of 28 August an air mass was transported through central Europe and arrived in Helsinki (Fig. 5a) carrying anthropogenic pollutants. A sudden drop in the concentrations of Ntot and $PM_1$ occurred at noon on 28 August when the wind turned and blew from the west until 10 AM on 30 August (Fig. 4c) over the Altantic and Baltic Sea carrying clean air with low particulate concentrations (Fig. 5b). Simultaneously, the diurnal variation of $CO_2$ diminished. During that period there was no precipitation. During the last 12 hours before the clean air mass arrived in Harmaja the average background particle concentrations stayed rather constant at $\sim 2.7 \times 10^3$ #/cm$^3$, whereas the $PM_1$ increased from $\sim 4$ µg/m$^3$ to $\sim 11$ µg/m$^3$. This indicates that also larger particles were transported from Europe. In fact, this is obvious from Fig. 6, which presents the average number size distribution (Fig. 6a) as well as the volume size distribution (Fig. 6b) of background particles in the evening of 27 August, at noon on 28 August just before the clean air mass arrived, in the afternoon of 28 August, and early in the morning of 1 September. In the last-mentioned case the particle number concentration was highest ($3.6 \times 10^3$ #/cm$^3$), but due to the small particle sizes (Fig. 6a) they did not have an effect on the volume (and mass) size distribution (Fig. 6b).

Note:

Because in the manuscript text all $CO_2$ fluxes by the EC method are given in the unit of $\mu mol\ m^{-2}\ s^{-1}$, we converted the median $F_{CO2}$ ($mg\ m^{-2}\ s^{-1}$) in Table 1 accordingly. We also found a typing error in its uncertainty value, which should be read 30.0%.